# Multifunctional lithium niobate platform for photodetection and photoacoustic and thermoelastic gas sensing

Haoyang Lin[1], Huadan Zheng[1] ✉, Wenguo Zhu[1], Yongchun Zhong[1], Jianhui Yu [1] ✉, Hongpeng Wu[2], Zhiwei Jia[3], Jinchuan Zhang [3], Angelo Sampaolo[2,4], Pietro Patimisco[2,4], Huihui Lu [1], Xiaojun Jia [2], Vincenzo Spagnolo [2,4] & Lei Dong [2] ✉

Leveraging the intrinsic multi physics nature of ferroelectric lithium niobate, we present a multi-functional platform (LN-MFP) that seamlessly integrates photoacoustic spectroscopy, light-induced thermoelastic spectroscopy and photodetection into a single on-chip device. The proposed LN-MFP operates over a broad spectral range spanning from the visible to the mid infrared. We experimentally demonstrate trace gas detection of nitrogen dioxide, water vapor, acetylene, carbon dioxide, methane and ammonia, achieving parts-per-billion detection limits. We implement a custom packaging solution where the LN-MFP chip and a 4.6 μm quantum cascade laser chip are mounted on a printed circuit board together with transimpedance amplification, demonstrating system-level integration. Using this co-packaged module, we demonstrate carbon monoxide detection via second-harmonic measurements, outlining a clear route towards fully integrated on-chip implementations. This compact, hybrid, multi-functional architecture markedly reduces system complexity and footprint compared with conventional benchtop systems and is intrinsically compatible with the rapidly developing lithium niobate integrated photonics ecosystem. The LN-MFP provides a core sensing building block for future all-lithium-niobate spectroscopic chips for environmental monitoring, point-of-care diagnostics and on-site chemical analysis.

Lithium niobate (LN) has emerged as a key material for on-chip devices, thanks to its outstanding electro-optic, piezoelectric, and nonlinear optical properties[1]. These unique characteristics make it a versatile platform for diverse integrated photonic applications[2]. Breakthroughs in LN photonics have propelled the development in high-performance components, including electro-optic modulators[3-5], frequency shifters[6], frequency combs[7-9], pulse generators[10], tunable lasers[11], spectroscopic sensing[12-14] and photonic integrated circuits[15]. Such advancements are critical enablers for optical communications, signal processing, and high-sensitivity sensing technologies[16,17]. Moreover, LN's distinctive piezoelectric and thermoelastic properties unlock new possibilities for high-performance acoustic and acousto-optic devices, expanding its range of applications[18-20]. The advancement of LN-based on-chip devices is pivotal to the future of integrated

[1]Key Laboratory of Optoelectronic Information and Sensing Technologies of Guangdong Higher Education Institutes, Department of Optoelectronic Engineering, Jinan University, Guangzhou, China. [2]State Key Laboratory of Quantum Optics and Quantum Optics Devices, Institute of Laser Spectroscopy, Shanxi University, Taiyuan, China. [3]Laboratory of Solid-State Optoelectronics Information Technology, Institute of Semiconductors, Chinese Academy of Sciences, Beijing, China. [4]PolySense Lab—Dipartimento Interateneo di Fisica, University and Politecnico of Bari, Bari, Italy. ✉e-mail: zhenghuadan@jnu.edu.cn; jianhuiyu@jnu.edu.cn; donglei@sxu.edu.cn

photonics and electromagnetic spectrum technologies[21]. LN's exceptional versatility positions it as a cornerstone for the next generation of compact, high-performance, and energy-efficient photonic systems.

Advancements in spectroscopic methodologies are expanding the frontiers photonics, offering unprecedented insights into molecular structures and dynamics. These breakthroughs are transforming fields such as chemistry, physics, and environmental science[22–30]. Among the spectroscopic methods, photoacoustic spectroscopy (PAS) stands out for its non-destructive nature and versatility across diverse type of samples over an extended range of wavelengths[31–35]. Light-induced thermoelastic spectroscopy (LITES) is an optical gas sensing technique that detects target gases by converting the sensor's thermoelastic deformation, induced by modulated light absorption, into electrical signals. This enables contact-based detection with high sensitivity and selectivity, particularly when using materials with excellent piezoelectric properties[36–38].

Despite their individual strengths, integrating spectroscopic detectors remains a major challenge for advancing micro-, nano- and on-chip spectroscopic devices. Conventional systems often rely on bulky optical components and complex configurations, limiting their suitability for miniaturized and portable sensing applications[39]. Key obstacles include large instrumentation, high power consumption, and the precise alignment of optical components, which hinder field deployment and real-time, on-site analysis. Furthermore, lack of seamless integration across multiple spectroscopic techniques restricts the development of versatile, compact platforms capable of delivering comprehensive analytical insights.

To overcome these challenges, this work introduces lithium niobate as a multi-functional integrated platform (LN-MFP) for spectroscopic sensing. By harnessing LN's strong piezoelectric and thermoelastic properties, we develop an integrated sensor capable of performing diverse spectroscopic measurements within a single device. The LN-MFP seamlessly combines photoacoustic, thermoelastic, and photodetection functionalities, addressing the limitations of conventional spectroscopic systems. This integration not only minimizes physical footprint and system complexity but also enhances sensitivity and selectivity through the synergistic interplay of multiple detection mechanisms. Incorporating resonance mechanisms can enhance sensor performance, improving both sensitivity and selectivity across diverse applications[40–42]. In this work, we leverage lithium niobate's piezoelectric properties and thermoelastic effects to detect and amplify signals via mechanical resonance, further optimized through a fork-shaped structure. This work has the potential to revolutionize on-chip spectroscopic devices by providing a compact, high-performance, and versatile solution that that overcomes the limitations of conventional bulky systems. The LN-MFP not only allows to advance the fundamental understanding of LN-based integrated platforms but also lays the groundwork for future innovations in the spectroscopic technologies. By addressing key integration challenges and enhancing spectroscopic detection capabilities, this research marks a pivotal step toward the next generation of multifunctional, on-chip spectroscopic sensors, driving progress in integrated photonics and advanced sensing technologies.

## Results

### Design and fabrication of the lithium niobate multi-functional platform

The design philosophy of the LN-MFP fundamentally differs from that of conventional quartz- or Si-based PAS/LITES devices. Lithium niobate is a ferroelectric crystal with 3 m point-group symmetry and spontaneous polarization, leading to piezoelectric coefficients and electromechanical coupling factors more than one order of magnitude larger than those of quartz (for example, $d_{22} \approx 25$ pC/N for LN versus $d_{11} \approx 2.3$ pC/N for quartz). At the same time, LN exhibits strong electro-optic,

nonlinear-optical, and pyroelectric effects that are absent in quartz and most MEMS materials. These intrinsic properties mean that LN-based sensors cannot be obtained by simply scaling or copying established quartz tuning fork designs; instead, the fork geometry, electrode layout, and fabrication process in this work are specifically engineered to harness LN's multi-physics coupling. As a result, a single LN-MFP chip can simultaneously operate as a high-efficiency acoustic transducer for PAS, a thermoelastic detector, and a platform for LITES, thereby enabling multimodal operation that is not available in conventional quartz or MEMS devices.

The design of the LN-MFP is shown in Fig. 1a. Simulations performed with COMSOL Multiphysics software provided a detailed analysis of the electrical response of lithium niobate under varying stress and thermal conditions. The simulation results in Fig. 1b, depict the charge distribution across the LN-MFP during vibration. Positive charge accumulations appear in the red regions, while negative charges are shown in the blue regions. Notably, negative charges are concentrated near the gap between the tines, whereas positive charges accumulate near the device's outer boundaries, forming a distinct bipolar distribution. This pattern suggests that mechanical deformation or thermal effects enhance the piezoelectric response, particularly in the marginal regions. Moreover, the symmetric charge distribution across both tines underscores their synchronized motion during vibration.

The LN-MFP is fabricated through an eight-step microfabrication process (Fig. 1c). It begins with slicing a y-cut 128° lithium niobate crystal block into thin wafers, each measuring 500 μm in thickness and 4 inches in diameter. The wafer edges are then refined using circular grinding to remove surfaces and ensure dimensional uniformity. Next, dual rotating pads polish the wafers to achieve a smooth, uniform finish. Finally, a spray-type cleaning system removes any contaminants and residues, preparing the wafer for subsequent processing. After cleaning, both sides of the wafers are coated with a 0.2 μm gold film. The wafers are then precisely diced into individual components. The structural design of the LN-MFP is fabricated using mechanical polishing and laser processing. A more specific and comprehensive description of the process flow was provided in Supplementary Note 3. The tines of the LN-MFP are ~ 11.5 mm in length and 1.7 mm in width, with a gap of 1 mm between them. The base has a width of $W \sim 7.6$ mm, and features a circular fillet with a radius of $r \sim 0.8$ mm at the junction between the base and the tines. The LN-MFP has a frequency of 10,485 Hz and a $Q$ factor of 1,621. An optical microscopy image of the fabricated LN-MFP is shown in Fig. 1d, while Fig. 1e and Fig. 1f present SEM images of the LN-MFP tine edge and the base fillet, respectively.

The design of the LN-MFP was guided by two primary factors: resonance frequency and the efficient coupling with laser sources. To minimize $1/f$ noise during spectroscopic detection, controlling the resonance frequency is crucial. In photoacoustic and photothermoelastic detection, the modulation frequency must be moderate, as certain molecular non-radiative relaxation processes can persist for up to 100 microseconds. Therefore, optimizing the resonance frequency is essential to maximizing device sensitivity while minimizing noise. Additionally, light sources like terahertz (THz) quantum cascade lasers (QCLs)[43] or Fabry-Perot diode lasers[44] often exhibit suboptimal beam quality. To accommodate these sources, the gap between the tines is intentionally widened. Microelectrodes were fabricated on both sides of the device using magnetron sputtering and masking techniques.

### Photoacoustic detection with the LN-MFP

To assess the photoacoustic detection capabilities of the LN-MFP, we employed light sources of different types covering a broad spectral range, from visible to long-wave infra-red wavelengths, specifically at 450 nm, 1.3 μm, 1.5 μm, 2 μm, 3.3 μm, and 9.77 μm, shown in the Fig. 2a.

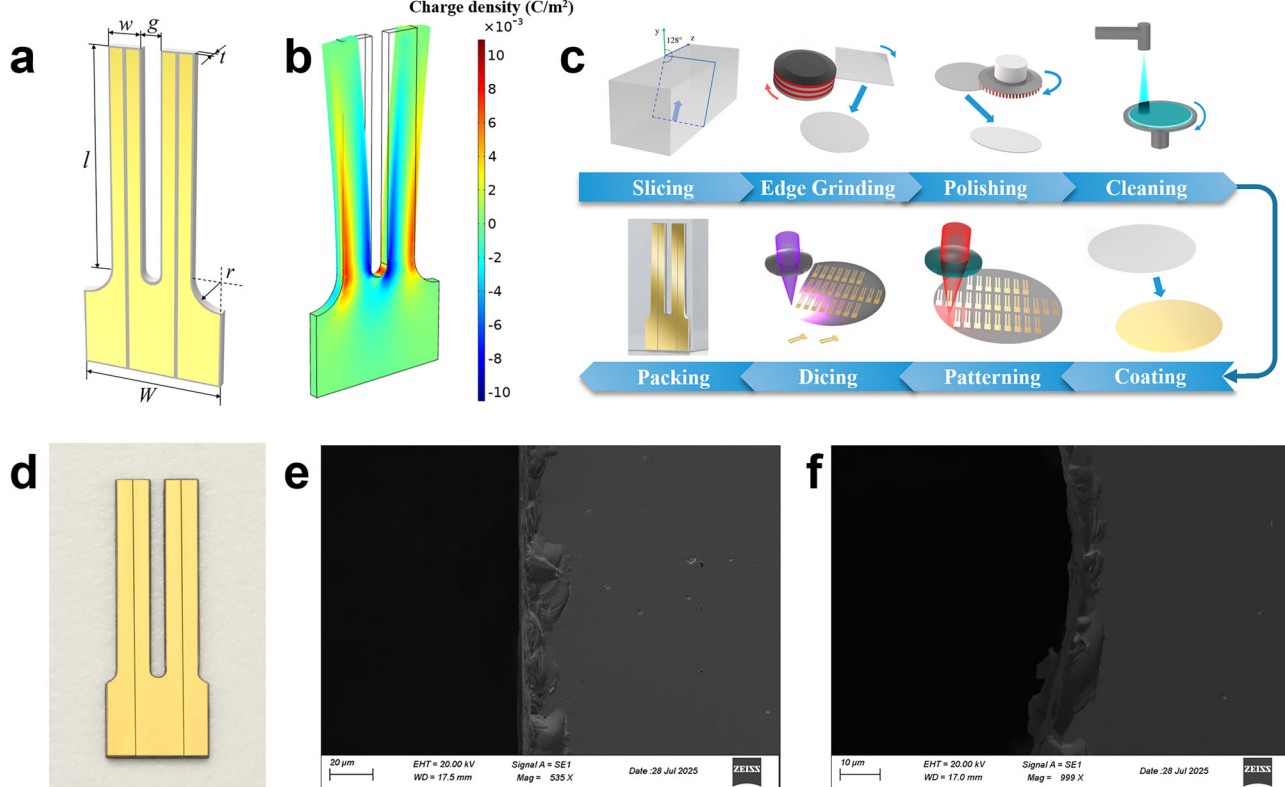

**Fig. 1 | Lithium Niobate Multi-Functional Platform. a** Design of the LN-MFP. **b** Simulated surface charge distribution of LN-MFP during vibration. The red regions correspond to positive charge accumulations, blue regions to negative ones. **c** Manufacturing process of LN-MFP; **d** Optical microscope image of the fabricated LN-MFP; **e** SEM image of the edge of the LN-MFP tine. **f** SEM image of the fillet of LN-MFP base.

This wide spectral operating range enables the detection of various gas species by targeting their characteristic absorption lines. All experiments were carried out at room temperature and pressure.

Nitrogen dioxide ($NO_2$) detection is crucial for environmental monitoring and public health protection. To measure $NO_2$ concentration, we used a visible (VIS) blue LED with a center wavelength of 450 nm as the light source. The target gas, supplied at a certified concentration of 5 ppm in $N_2$, was analysed, and the results are shown in Fig. 2b. Since the LED operated at a fixed wavelength, intensity modulation was applied to excite $NO_2$ molecules and generate acoustic waves. The LED emitted an output power of ~1.2 W, operating at a modulation frequency of 10,485 Hz with a 50% duty cycle. The LN-MFP achieved a signal-to-noise ratio (SNR) of ~132, corresponding to a minimum detection limit (MDL) of ~38 ppb.

The concentration of water vapor ($H_2O$) in air was measured using a 1.3 μm near-infrared (NIR) distributed feedback (DFB) semiconductor laser, with the results presented in Fig. 2c. The laser wavelength was scanned from 7185.2 $cm^{-1}$ to 7186.1 $cm^{-1}$, covering the water absorption line at 7185.6 $cm^{-1}$. At a water vapor concentration of 18,000 ppm, the peak voltage of the second harmonic signal ($2f$) was recorded at 0.0618 V. The 1σ standard deviation at a non-absorption wavelength was determined to be $3.65 \times 10^{-6}$ V, resulting in a $H_2O$ MDL of ~1 ppm.

The detection of acetylene ($C_2H_2$) is essential for ensuring safety in industrial processes and monitoring human activities. Acetylene at a concentration of 50 ppm is detected in $N_2$ using a 1.5 μm near-infrared semiconductor laser (Fig. 2d). The laser wavelength was scanned from 6506.99 $cm^{-1}$ to 6507.63 $cm^{-1}$, covering the $C_2H_2$ absorption line at 6507.4 $cm^{-1}$. The peak voltage of the $2f$ signal was recorded at 0.0477 V, while the modulation depth was to 0.44 $cm^{-1}$. As a result, the 1σ standard deviation at a non-absorption wavelength was $5.99 \times 10^{-6}$ V, yielding a MDL of ~63 ppb for $C_2H_2$.

Carbon dioxide ($CO_2$), a greenhouse gas closely linked to climate change, was detected using a 2 μm NIR DFB laser diode. A certified $CO_2$ gas mixture with a concentration of 1000 ppm in $N_2$ was analyzed, as shown in Fig. 2e. The laser wavelength was scanned from 4988.53 $cm^{-1}$ to 4988.77 $cm^{-1}$, covering the $CO_2$ absorption line at 4988.65 $cm^{-1}$. The $2f$ peak signal was recorded at $1.34 \times 10^{-3}$ V, with a 1σ standard deviation of $1.14 \times 10^{-5}$ V measured away from the absorption line. These results indicate that the LN-MFP achieved a MDL of ~9 ppm for $CO_2$.

Methane ($CH_4$), a major greenhouse gas and widely used energy source, was analyzed using a 3.3 μm mid-wave infrared (MWIR) interband cascade laser (ICL) in combination with the LN-MFP. The results are shown in Fig. 2f. The ICL wavelength was scanned from 2967.86 $cm^{-1}$ to 2969.33 $cm^{-1}$, covering multiple methane absorption lines. Methane exhibits four absorption lines at 2968.86 $cm^{-1}$, 2968.73 $cm^{-1}$, 2968.47 $cm^{-1}$, and 2968.4 $cm^{-1}$. A certified methane gas sample with a concentration of 100 ppm in $N_2$ was used for the measurement. At the strongest absorption peak, the signal voltage reached 0.019 V, with a 1σ standard deviation of $9.4 \times 10^{-6}$ V measured away from the methane absorption lines. This corresponds to a MDL of ~49 ppb.

Ammonia ($NH_3$), a critical compound in both industrial and agricultural applications, was analyzed using a 9.77 μm long-wave infrared (LWIR) QCL. The results are shown in Fig. 2g. The laser wavelength was tuned from 1022.4 $cm^{-1}$ to 1023 $cm^{-1}$, covering the ammonia absorption line at 1022.76 $cm^{-1}$. A certified ammonia gas sample with a concentration of 1 ppm in $N_2$ was used for the measurement. The system achieved a SNR of ~145, corresponding to an MDL of ~7 ppb for $NH_3$.

Responses to varying target-gas concentrations were measured and the corresponding calibration curves are plotted in Fig. 2. As shown in Fig. 2, all channels exhibit excellent linearity, with linear-fit

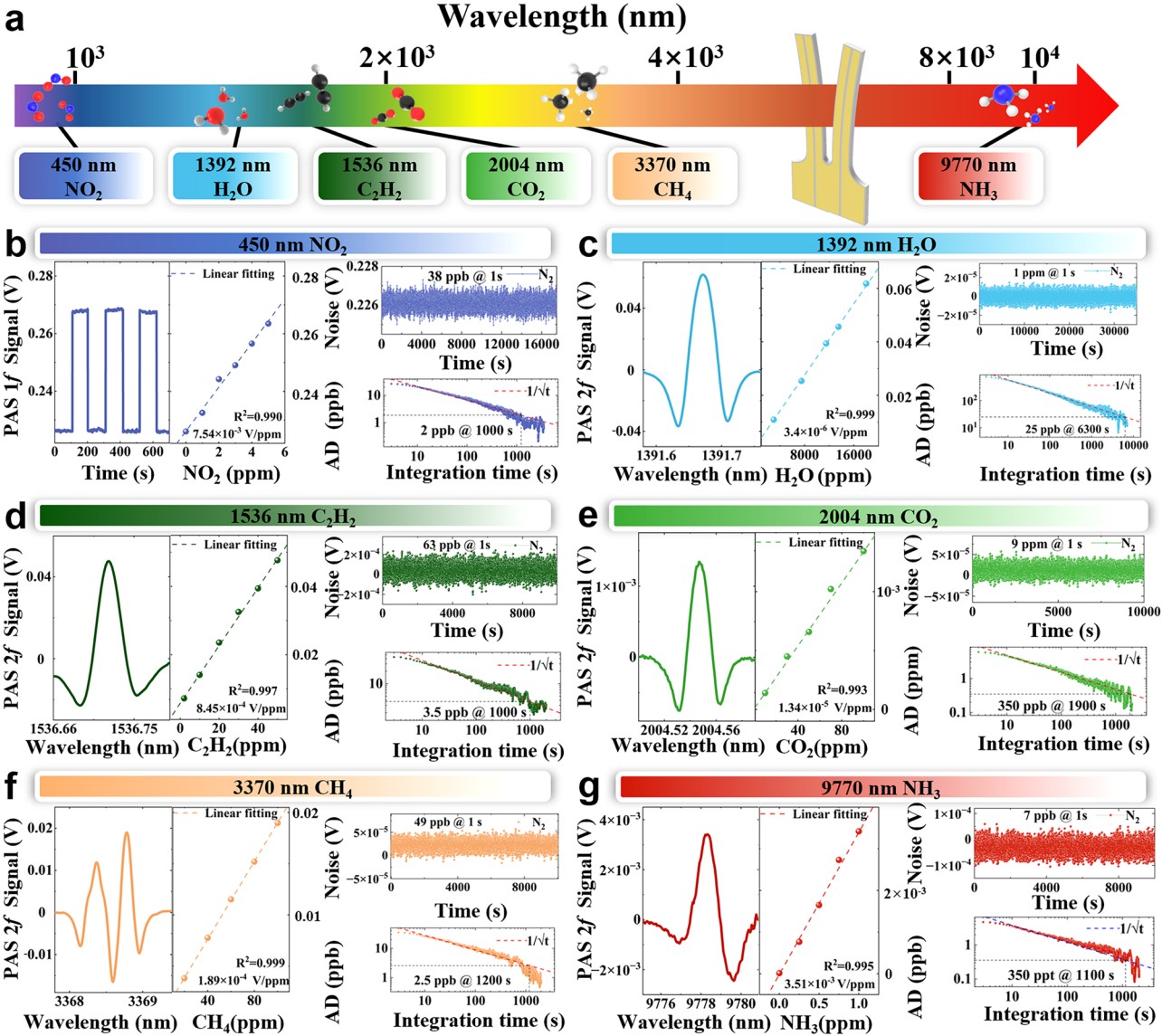

**Fig. 2 | LN-MFP for photoacoustic detection. a** Spectral coverage with excitation sources from visible to long-wave infrared and the targeted analytes. **b**–**g** present representative measurements for each target gas: the main plot in each panel shows a typical $2f$ signal acquired with the indicated light source and sample concentration; the left inset shows the concentration-calibration (linear fit); the top-right inset shows a $1\sigma$ noise level obtained with pure $N_2$; and the bottom-right inset shows the Allan-deviation (AD) curves used to extract the $1\sigma$ MDL and optimal averaging times. Extracted MDLs ($1\sigma$) and averaging times are: **b** MDL - 2 ppb at 1,000 s; **c** MDL - 25 ppb at 6,300 s; **d** MDL - 3.5 ppb at 1000 s; **e** MDL - 350 ppb at ~1900 s; **f** MDL - 2.5 ppb at 1,200 s; **g** MDL - 350 ppt at 1100 s. VIS LED: visible light emitting diode; NIR LD: near-infrared laser diode; MWIR ICL: mid-wave interband cascade laser; LWIR QCL: long-wave quantum-cascade laser.

coefficients of $R^2 > 0.99$. To assess the long-term stability of the LN-MFP PAS sensor, we performed noise characterization and Allan-deviation analysis (Fig. 2). The $1\sigma$ noise levels were measured under a continuous flow of 200 sccm pure $N_2$. Figure 2b–g show the corresponding noise traces and Allan-deviation curves for the LN-MFP operated with a 450 nm LED, 1392 nm laser diode, 1536 nm laser diode, 2004 nm laser diode, 3370 nm ICL, and 9770 nm QCL. From the Allan deviation analysis, the $1\sigma$ MDLs, and optimal averaging times were extracted: the 450 nm ($NO_2$) channel achieves an MDL of ~2 ppb at an averaging time of 1000 s; the 1392 nm ($H_2O$), ~25 ppb at 6300 s; the 1536 nm ($C_2H_2$), ~3.5 ppb at 1000 s; the 2004 nm ($CO_2$), ~350 ppb at 1900 s; the 3370 nm ($CH_4$), ~2.5 ppb at 1200 s; and the 9770 nm ($NH_3$), ~350 ppt at 1000 s.

Table 1 provides summary of the sensitivities achieved together with the calculated normalized noise equivalent absorption coefficient (NNEA) values and the optimal modulation amplitude. The photoacoustic detection performance of the LN-MFP and the QTF-based PAS

systems is summarized in Supplementary Note 10. These results demonstrate the LN-MFP's capability to detect multiple gas species across a broad spectral range with high sensitivity, underscoring its versatility and potential for practical applications not limited to environmental monitoring and industrial process control.

**Photodetection with the LN-MFP**

Beyond its capabilities in photoacoustic detection, the LN-MFP demonstrates remarkable potential for photodetection across a broad spectral range, from visible light at 450 nm to long-wave infrared radiation at nearly 10 µm. To assess its photodetection performance, square-wave modulated lasers at various wavelengths were shined onto the LN-MFP most sensitive area. The modulated light intensity was gradually reduced to evaluate the device's response, enabling the determination of its minimum detectable power and overall detectivity. The response curves of the LN-MFP were measured for light wavelength ranging from 450 nm to 9.77 µm (Fig. 3).

**Table. 1 | Summary of LN-MFP photoacoustic detection performance**

| Light sources | Wavelength (µm) | Molecules | Power (mW) | Modulation amplitude | Sensitivity | NNEA (cm⁻¹ W Hz⁻¹ᐟ²) |
|---|---|---|---|---|---|---|
| VIS LED | 0.45 | $NO_2$ | 1200 | / | 38 ppb | / |
| NIR LD | 1.39 | $H_2O$ | 19 | 0.86 cm⁻¹ | 1 ppm | $5.82 \times 10^{-9}$ |
| NIR LD | 1.53 | $C_2H_2$ | 3700 | 0.44 cm⁻¹ | 63 ppb | $2.7 \times 10^{-8}$ |
| NIR LD | 2 | $CO_2$ | 10.6 | 0.27 cm⁻¹ | 9 ppm | $2.53 \times 10^{-8}$ |
| MWIR ICL | 3.37 | $CH_4$ | 6.5 | 0.29 cm⁻¹ | 49 ppb | $8.36 \times 10^{-9}$ |
| LWIR QCL | 9.77 | $NH_3$ | 82.3 | 0.14 cm⁻¹ | 7 ppb | $6.58 \times 10^{-10}$ |

VIS LED visible light emitting diode, NIR LD, MWIR ICL mid-wave infrared interband cascade laser, LWIR QCL long-wave infrared quantum cascade lasers. NNEA normalized noise equivalent absorption coefficient.

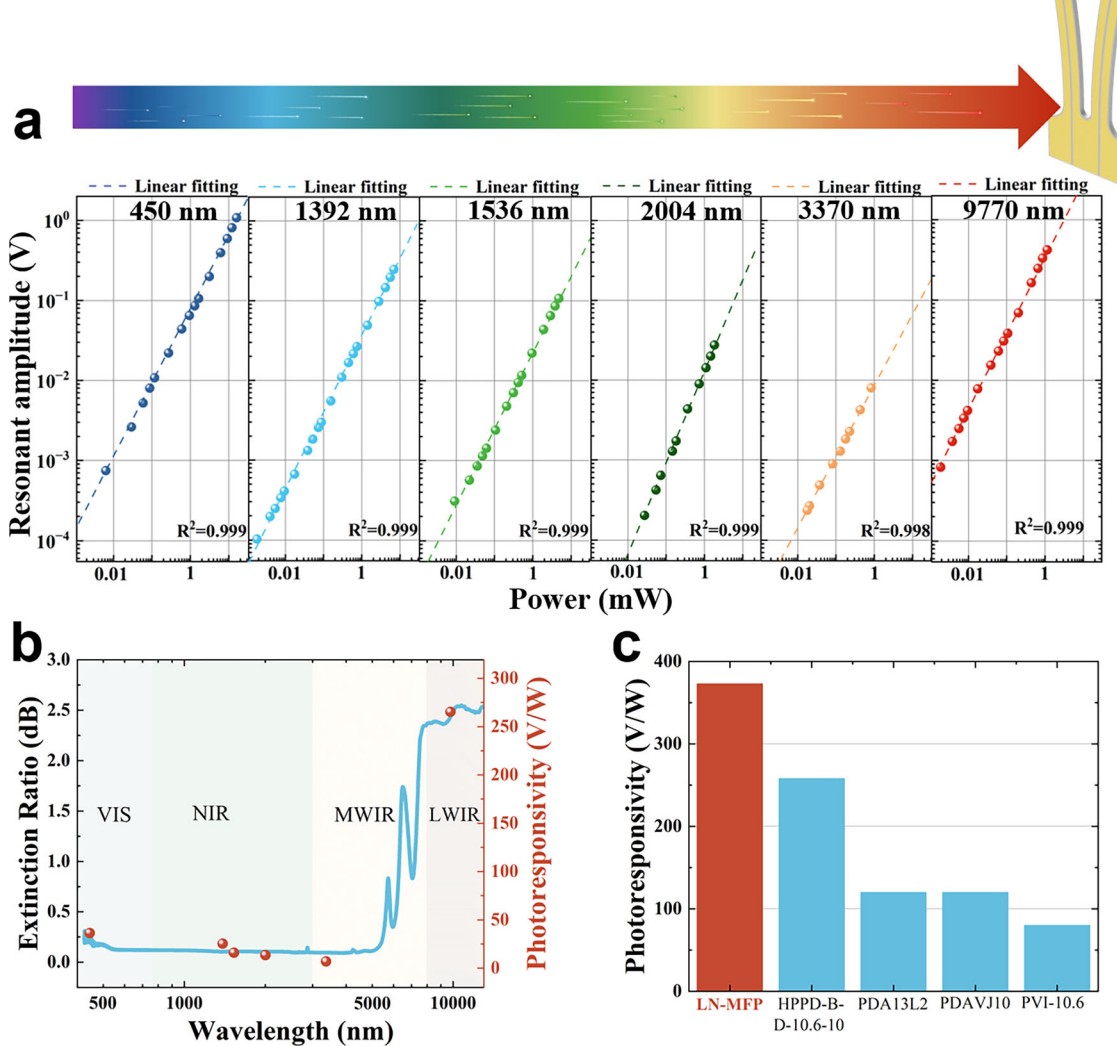

**Fig. 3 | LN-MFP for photodetection. a** Resonance amplitude of LN-MFP to 450 nm VIS LED, 1.39 µm NIR LD, 1.53 µm NIR LD, 2 µm NIR LD, 3.37 µm MWIR ICL, and 9.77 µm LWIR QCL. **b** The LN-MFP photoresponsivity to wavelengths from VIS to LWIR, compared to the FTIR absorption spectrum of a LN wafer. **c** Photoresponsivity of LN-MFP compared with four commercial detectors at 9.7 µm, including Healthy Photo infrared detector model HPPD-B-D-10.6-10

(HgCdTe), Thorlabs pyroelectric detector model PDA13L2 (LiTaO3), photoelectric detector model PDAVJ10 (HgCdTe), Vigo infrared detector model PVI-10.6 (HgCdTe). The results of LN-MFP and HPPD-B-D-10.6-10 were obtained by experiments. The results of PDA13L2, PDAVJ10, and PVI-10.6 were obtained from the datasheet.

For the 450 nm LED as the modulation depth decreased from 0.904 cm⁻¹ to $3.57 \times 10^{-4}$ cm⁻¹, the vibration amplitude of the LN-MFP correspondingly dropped from 1.08 V to $7.47 \times 10^{-4}$ V. Using Equation (1) in the Supplementary Note 7, the sensitivity of the LN-MFP to 450 nm light was determined to be 260 nW, with a corresponding

detectivity of 67.7 V/W. The diode laser operating at 1.3 µm was detuned from the $H_2O$ absorption peak to avoid any power absorption occurring along the optical pathlength between laser and photodetector. For the 1.3 µm wavelength, as the modulation depth decreased from 1.15 cm⁻¹ to $2.98 \times 10^{-4}$ cm⁻¹, the vibration amplitude

**Table. 2 | LN-MFP for photodetection from VIS to LWIR**

| Detector type | Light sources | Wavelength (μm) | MDL (nW) | NEP (W/√Hz) | Photoresponsivity (V/W) |
|---|---|---|---|---|---|
| LN-MFP | VIS LED | 0.45 | 260 | $5.88 \times 10^{-7}$ | 67.7 |
| LN-MFP | NIR LD | 1.39 | 445 | $1 \times 10^{-6}$ | 35.5 |
| LN-MFP | NIR LD | 1.53 | 695 | $1.57 \times 10^{-6}$ | 22.5 |
| LN-MFP | NIR LD | 2 | 693 | $1.56 \times 10^{-6}$ | 15.3 |
| LN-MFP | MIR ICL | 3.37 | 1000 | $2.26 \times 10^{-6}$ | 9.8 |
| LN-MFP | LWIR QCL | 9.77 | 25 | $5.65 \times 10^{-8}$ | 373 |
| HPPD-B-D-10.6-10 | LWIR QCL | 9.77 | 8000 | $1.8 \times 10^{-5}$ | 258 |

LN-MFP lithium niobate as a multi-functional integrated platform, HPPD-B-D-10.6-10, Healthy Photo infrared detector model (HgCdTe), VIS LED visible light emitting diode, NIR LD, MWIR ICL mid-wave infrared interband cascade laser, LWIR QCL long-wave infrared quantum cascade lasers, MDL minimum detection limit, NEP noise equivalent power.

correspondingly dropped from $2.47 \times 10^{-1}$ V to $1.03 \times 10^{-4}$ V. The sensitivity was determined to be 445 nW, with a detectivity of 35.5 V/W. Similarly, for the 1.53 μm wavelength, decreasing the modulation depth from 0.59 cm$^{-1}$ to $1.12 \times 10^{-3}$ cm$^{-1}$ led to a reduction in vibration amplitude from $1.07 \times 10^{-1}$ V to $3.1 \times 10^{-4}$ V. The sensitivity of the LN-MFP to the 1.53 μm laser was then determined to be 695 nW, with a corresponding detectivity of 22.5 V/W. For the 2 μm NIR semiconductor laser, when the modulation depth was lowered from 0.33 cm$^{-1}$ to $4.95 \times 10^{-3}$ cm$^{-1}$, the vibration amplitude simultaneously decreased from $2.76 \times 10^{-2}$ V to $2.02 \times 10^{-4}$ V. Under these conditions, the sensitivity was measured at 693 nW, and the detectivity reached 15.3 V/W. At 3.37 μm the laser modulation depth was reduced from 0.34 cm$^{-1}$ to $7.22 \times 10^{-3}$ cm$^{-1}$ and the vibration amplitude decreased from $8.03 \times 10^{-3}$ V to $2.35 \times 10^{-4}$ V. The sensitivity was calculated to be 1 μW, with a corresponding detectivity of 9.8 V/W. Finally, at 9.77 μm as the modulation depth decreased from $2.47 \times 10^{-2}$ cm$^{-1}$ to $8.25 \times 10^{-5}$ cm$^{-1}$, the vibration amplitude dropped from 0.42 V to $1.73 \times 10^{-3}$ V. The sensitivity to the 9.77 μm laser was determined to be 25 nW, with a detectivity of $3.73 \times 10^{2}$ V/W.

The performance of the LN-MFP is comprehensively assessed across the VIS, NIR, MWIR and LWIR spectral regions, spanning wavelengths from 450 nm to 9.77 μm (Fig. 3b). The results show that the photoresponsivity initially decreases as the wavelength transitions from the VIS to the LWIR region, followed by an increase in the LWIR region. This trend indicates that the LN-MFP achieves its highest sensitivity in the LWIR range. This behavior is attributed to the intrinsic optical absorption properties of LN in the investigated range, as illustrated in Fig. 3b, which also includes the absorption spectrum of LN, which was obtained by measuring the absorption of a 500 μm thick LN wafer using Fourier transform infrared spectroscopy. The enhanced absorption in the MIR and LWIR regions further underscores the LN-MFP's potential for high-sensitivity applications, such as the fundamental absorption band of molecules. Figure 3c compares the photoresponsivity of the LN-MFP to that of four commercial mid-infrared detectors in response to a 9.7 μm laser. Notably, the LN-MFP exhibits superior photoresponsivity at 9.7 μm, outperforming these commercial mid-infrared photodetectors.

Table 2 summarizes the LN-MFP's photodetection performance across various wavelengths. All measurements were conducted at room temperature without active cooling. Notably, the MDL and response rate could be significantly enhanced by incorporating cooling strategies, as lower temperatures generally reduce thermal noise and improve detector performance. This suggests that further optimization through thermal management could unlock even greater sensitivity for advanced spectroscopic applications.

## Thermoelastic detection with the LN-MFP
Beyond its photoacoustic detection capabilities, the LN-MFP also exhibits strong potential for LITES. In this mechanism, target molecules absorb modulated laser radiation, producing localized heating and subsequent thermal expansion of a piezoelectric sensing element, in our case, the LN-MFP. This resulting mechanical deformation, enhanced by the LN-MFP's inherent mechanical resonance, excites a piezoelectric response, generating a measurable electrical signal. Notably, this resonant amplification effect boosts the sensitivity of the system.

To evaluate the LITES performance of the LN-MFP system, six different light sources spanning from VIS to LWIR were employed to analyze different gases molecules and the results are shown in Fig. 4a. The laser beams propagated in free space, with the LN-MFP positioned at an optical path length of 2.5 cm. All experiments were conducted under laboratory conditions at ambient temperature and pressure, with $N_2$ as the carrier gas.

A 450 nm visible LED was used to detect $NO_2$. The $1f$ signal was obtained by alternately introducing pure $N_2$ and 5% $NO_2$, as shown in Fig. 4b. The modulation depth was set to ~100%, with a duty cycle of 50%. The LED output power was set as 330 mW. By analyzing the signals for pure $N_2$ and 5% $NO_2$, the LN-MFP system achieved a SNR of 3692, corresponding to a MDL of ~ 13.5 ppm.

For the evaluation of the thermoelastic detection capability of the LN-MFP in the near- and mid-IR regions, the noise level was determined as the 1σ standard deviation at non-absorbing wavelengths.

At 1.3 μm a near-infrared distributed feedback (DFB) laser was employed to target the $H_2O$ absorption line at 7185.6 cm$^{-1}$, with the concentration fixed at 2% via a humidity controller. Under an optimized modulation depth of 0.4 cm$^{-1}$, the LN-MFP system generated a $2f$ LITES signal peak of 1.62 mV, as shown in Fig. 4c, yielding an MDL of ~59.3 ppm for $H_2O$. Similarly, a 1.5 μm near-infrared telecommunication laser was employed to detect $C_2H_2$ by targeting its absorption line at 6534.6 cm$^{-1}$. With the laser power set to 10.5 mW and a modulation depth of 0.68 cm$^{-1}$, the LN-MFP produced a $2f$ signal peak of 4.4 mV, while the noise resulted 9.16 μV, leading to an MDL of 10.4 ppm for $C_2H_2$ as depicted in Fig. 4d.

A 2 μm laser diode was employed to $CO_2$ at a 20% concentration. The laser was precisely tuned to target the $CO_2$ absorption line at 4988.65 cm$^{-1}$, and with a modulation amplitude of 0.47 cm$^{-1}$, the LN-MFP system generated a $2f$ signal peak of 3.1 mV, shown in the Fig. 4e. The baseline noise was determined to be $1.1 \times 10^{-5}$ V, corresponding to a MDL of ~ 680 ppm for $CO_2$. At 3.3 μm, a mid-wave infrared (MWIR) interband cascade laser (ICL) at was used to measure $CH_4$ at a concentration of 2%. The combined contribution of four $CH_4$ absorption lines at 2968.86 cm$^{-1}$, 2968.73 cm$^{-1}$, 2968.47 cm$^{-1}$ and 2968.40 cm$^{-1}$ produces a $2f$ signal, with the modulation depth optimized to 0.27 cm$^{-1}$, yielding a peak amplitude of 0.101 V (Fig. 4f). The 1σ noise level was determined to be 112 μV, corresponding to a MDL of ~ 22 ppm for $CH_4$.

Finally to evaluate the thermoelastic detection capability of the LN-MFP at long-wave infrared region, a 9.7 μm quantum cascade laser (QCL) was employed for ammonia ($NH_3$) detection. The QCL was tuned from 1022.4 cm$^{-1}$ to 1023 cm$^{-1}$, covering the $NH_3$ absorption line at

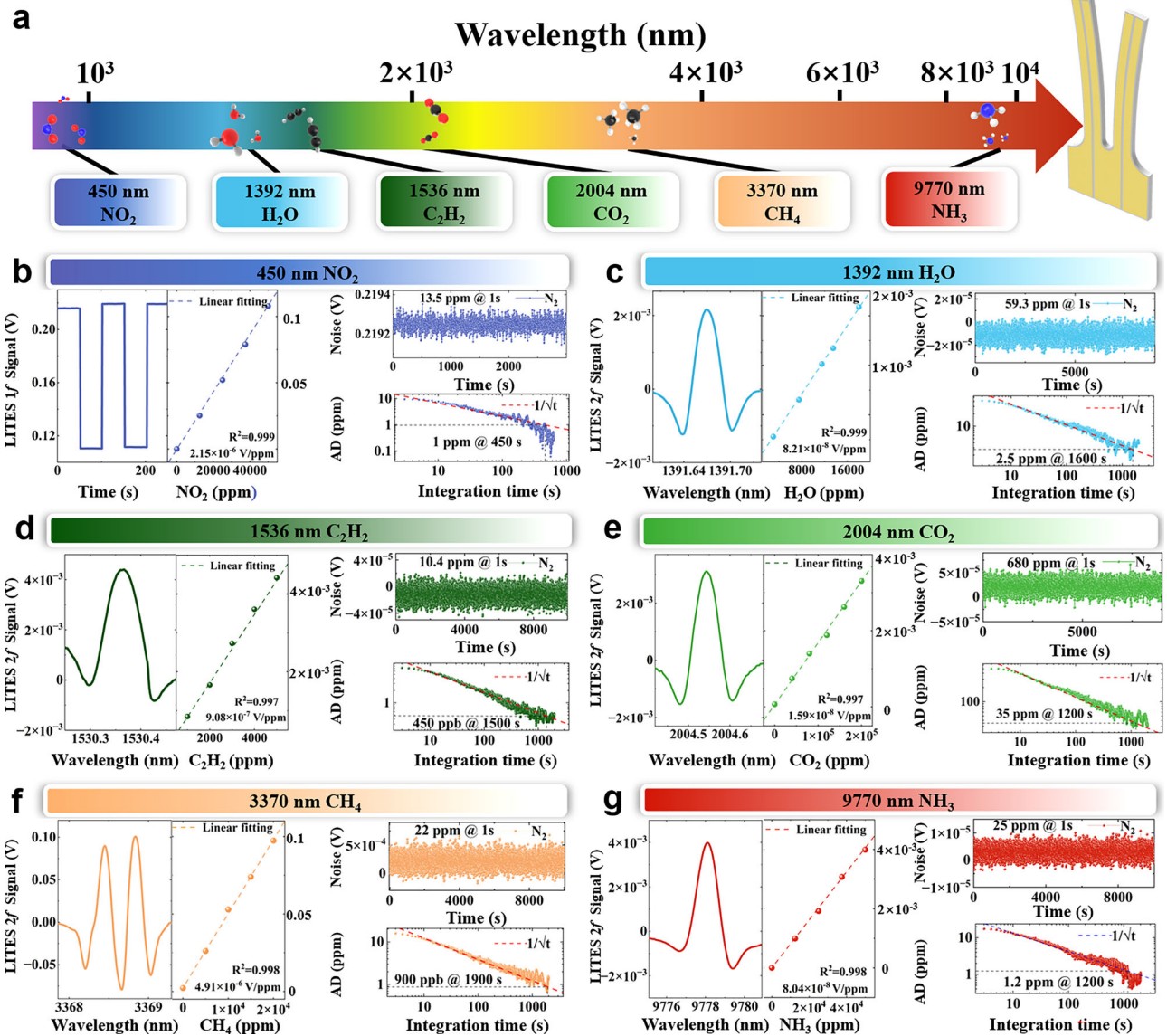

**Fig. 4 | LN-MFP for thermoelastic detection. a** Spectral coverage of excitation sources from visible to long-wave infrared, together with the corresponding targeted analytes. **b–g** Representative measurements for each target gas species: the main plots show typical 2*f* signal acquired with the indicated light source and gas concentration. The left inset displays the concentration-calibration (linear fit); the top-right insets show the 1σ noise level measured for pure $N_2$; and the bottom-right insets present the Allan-deviation (AD) curves used to extract the 1σ MDL and optimal averaging times. Extracted MDLs (1σ) and optimal averaging times are: **b** MDL - 1 ppm at 450 s; **c** MDL - 2.5 ppm at 1600 s; **d** MDL - 450 ppb at 1500 s; **e** MDL - 35 ppm at 1200 s; **f** MDL - 900 ppb at 1900 s; **g** MDL - 1.2 ppm at 1200 s.

1022.76 cm$^{-1}$. With an optimized modulation depth of -0.15 cm$^{-1}$ for a 5% $NH_3$ sample, the LN-MFP achieved a SNR of 2014, yielding a MDL of -25 ppm. The results were shown in Fig. 4g.

Similar to the PAS, calibration measurements for the LITES channels are summarized in Fig. 4. All channels exhibit excellent linearity with linear-fit R$^2$ > 0.99. To assess the long-term stability of the LN-MFP LITES sensor, we performed measurements identical to those used for the PAS characterization and computed Allan-deviation curves for each detection channel. From the Allan deviation analysis, the 1σ MDLs, and the corresponding optimal averaging times were extracted. The 450 nm ($NO_2$) channel achieves an MDL of -1 ppm at an optimal averaging time of 450 s; the 1,392 nm ($H_2O$) channel, -2.5 ppm at 1600 s; the 1536 nm ($C_2H_2$) channel, -450 ppb at 1500 s; the 2004 nm ($CO_2$) channel, -35 ppm at 1200 s; the 3370 nm ($CH_4$) channel, -900 ppb at 1900 s; and the 9770 nm ($NH_3$) channel -1.2 ppm at 1200 s.

Table 3 provides an overview of the LN-MFP's thermoelastic detection performance for various gas species across the range 450 nm - 9.77 μm. The table details the light sources type, the target molecules, the laser power, the modulation amplitude, the sensitivity, and the normalized noise-equivalent absorption (NNEA) achieved. Note that we implemented a very short optical path of only 2.5 cm and the sensitivity and NNEA could be simply improved by employing multi-pass cells or resonant cavities. A comparison with state-of-the-art LITES is provided in the Supplementary Note 10.

The integration of thermoelastic detection into the LN-MFP demonstrates its versatility as a multifunctional spectroscopic platform. The spectral selectivity of both modes arises from the intrinsic ro-vibrational fingerprints of target molecules. The dual functionality does not alter this selectivity but extends its applicability across varying gases and conditions. By incorporating both PAS and LITES detection modes, the device enables adaptive gas sensing across a

**Table. 3 | Summary of LN-MFP thermoelastic detection performance**

| Light sources | Wavelength (μm) | Molecules | Power (mW) | Modulation amplitude | Sensitivity (ppm) | NNEA (cm⁻¹ W Hz⁻¹/²) |
|---|---|---|---|---|---|---|
| VIS LED | 0.45 | $NO_2$ | 330 | / | 13.5 | / |
| NIR LD | 1.39 | $H_2O$ | 19 | 0.4 cm⁻¹ | 59.3 | $3.45 \times 10^{-7}$ |
| NIR LD | 1.53 | $C_2H_2$ | 9.8 | 0.68 cm⁻¹ | 10.4 | $2.33 \times 10^{-7}$ |
| NIR LD | 2 | $CO_2$ | 8.5 | 0.47 cm⁻¹ | 680 | $2 \times 10^{-6}$ |
| MWIR ICL | 3.37 | $CH_4$ | 6.5 | 0.27 cm⁻¹ | 22 | $3.75 \times 10^{-6}$ |
| LWIR QCL | 9.77 | $NH_3$ | 82.3 | 0.15 cm⁻¹ | 25 | $2.35 \times 10^{-6}$ |

VIS LED visible light emitting diode, NIR LD, MWIR ICL mid-wave infrared interband cascade laser, LWIR QCL long-wave infrared quantum cascade lasers. NNEA normalized noise equivalent absorption coefficient.

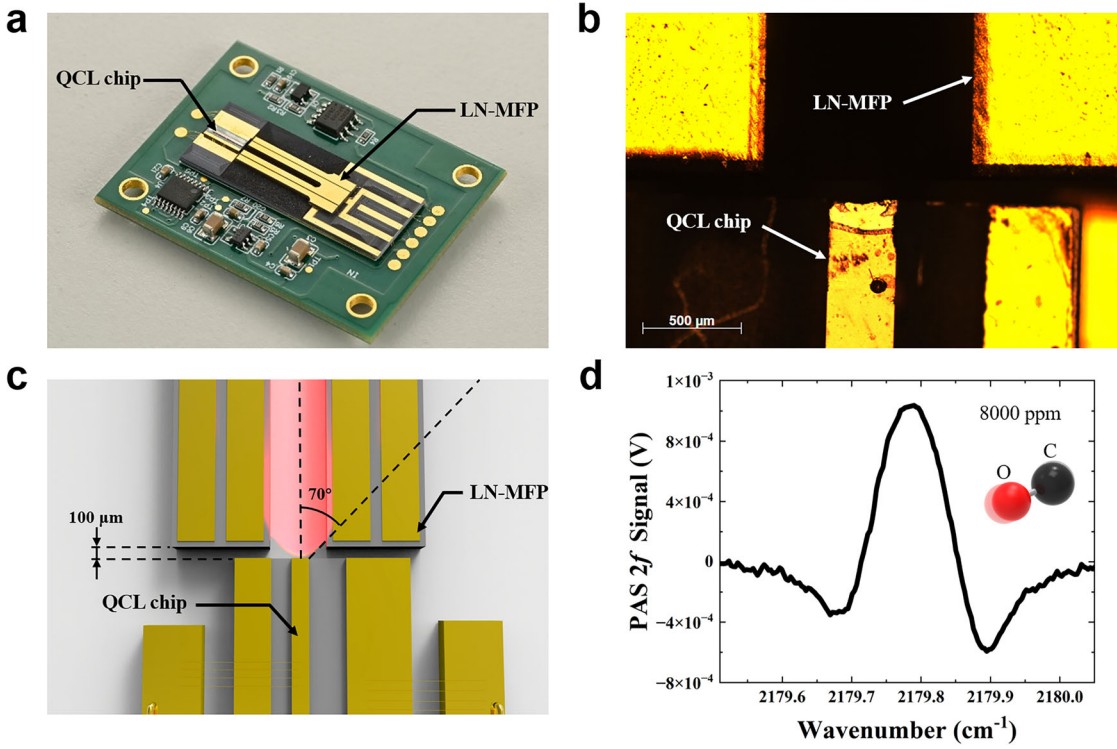

**Fig. 5 | PCB-level co-packaged LN-MFP-QCL sensor module. a** Photograph of the packaged PCB-level hybrid sensor module. **b** Optical-microscope image of the coupling region showing the QCL facet and the LN-MFP tine gap. **c** Schematic diagram of the free-space optical coupling. **d** 2*f* photoacoustic spectrum recorded from 8,000 ppm CO in $N_2$.

wide range of environments and measurement scenarios. In PAS mode, direct gas–sensor interaction provides high-sensitivity detection within a compact volume, ideal for confined or small-scale applications. Conversely, LITES operates in a non-contact configuration, suitable for sealed, corrosive, or otherwise harsh environments. This dual-mode architecture combines the sensitivity and compactness of contact-based detection with the robustness and flexibility of non-contact operation, enabling comprehensive spectroscopic analysis and positioning the LN-MFP as a versatile platform for next-generation sensing technologies.

## PCB-level Co-packaged LN-MFP Sensor Module

Based on the abovementioned multifunctional capabilities, we next implemented a PCB-level co-packaged LN-MFP sensor module and evaluated its performance. As shown in the photograph in Fig. 5a, the wire-bonded QCL chip and LN-MFP are co-mounted on a common silicon carrier and installed on a custom 3.5 cm × 4.5 cm PCB that integrates the drive and readout electronics, including a transimpedance amplifier to pre-condition the LN-MFP charge output. An optical-microscope close-up of the coupling between the QCL facet

and the LN-MFP tine gap is shown in Fig. 5b. The PCB features a stepped electrode pad to interface with the bottom-side electrode of the LN-MFP, while the top electrode is wire-bonded for signal extraction.

As proof of concept experiment, a QCL chip with the central wavelength of 4.6 μm was used. The output wavelength of this QCL chip was tuned by injection current to cover the CO absorption line at 2179.77 cm⁻¹. The LN-MFP is coupled to the QCL chip with a distance of ~100 μm. No collimating optics were used in the system. The optical coupling efficiency was estimated to be ~29% (Fig. 5b and c). We have initially optimized the modulation amplitude to 0.36 cm⁻¹. We employed a trumpet-shaped gas interface that directly covers the chip. The gas is fed to the sensor through this interface, flowing onto the chip at a rate of 100 sccm, consistent with standard experimental conditions, to help minimize noise. The 2*f* signal enables detection of 8000 ppm CO in $N_2$ using a PCB-level co-packaged LN-MFP–QCL module (Fig. 5d). This result demonstrates that the LN-MFP can serve as a compact and efficient sensing element in a hybrid QCL-LN configuration and highlighting a realistic pathway toward future fully integrated on-chip spectroscopic systems.

Furthermore, by co-packaging the LN-MFP chip with a mid-infrared QCL chip and the associated drive and readout electronics on a compact PCB, we demonstrate that the LN-MFP can be seamlessly interfaced with semiconductor laser sources using standard microelectronic assembly techniques. Experimental results reveal high sensitivity and selectivity across a broad spectral range, achieving detection limits suitable for real-world applications in environmental monitoring and industrial process control. Even though the current implementation is at the PCB level rather than monolithic, this hybrid QCL–LN architecture already reduces the footprint and alignment complexity of the spectrometer and establishes the LN-MFP as a practical building block for future fully integrated lithium-niobate spectroscopic chips.

## Discussion

In summary, we have established a lithium-niobate-based multi-functional platform (LN-MFP) that unifies three fundamentally different sensing mechanisms, photoacoustic spectroscopy, light-induced thermoelastic spectroscopy, and thermoelastic photodetection, within a single ferroelectric crystal chip. The LN-MFP exploits the strong piezoelectric, pyroelectric, electro-optic, and photoelastic responses of lithium niobate to achieve efficient acoustic-to-electric and thermoelastic transduction, while operating over a spectral range extending from the visible to the mid-infrared. Beyond the device level, we demonstrate a PCB-level co-packaged LN-MFP–QCL module capable of mid-infrared gas sensing, thereby validating that the LN-MFP can serve as a practical sensing front-end compatible with semiconductor laser sources and standard microelectronic assembly. Together with progress in LN-based lasers, frequency combs, and high-speed modulators, this multifunctional sensing platform fills a key gap in the LN photonics landscape and points toward future all-lithium-niobate spectroscopic chips in which light generation, modulation, and multimodal detection are realized within a single material system. We therefore view the main advance of this work not as an incremental improvement in a single performance metric, but as the introduction of a new LN-centric material and system-level paradigm for integrated spectroscopic sensing.

In future work, we aim to achieve full photonic integration by combining the LN-MFP with an on-chip light source to develop a completely integrated optical sensor chip. Specifically, our goal is to integrate a mid-infrared quantum cascade laser (QCL) chip and optical waveguide with the LN-MFP through advanced microfabrication techniques, thereby unifying the light source, waveguide, and detector on a single monolithic platform. This seamless integration will enable simultaneous mass production of the entire system on one chip, further miniaturizing the sensor and enhancing its functionality. Such a breakthrough is expected to open new avenues for applications in point-of-care diagnostics, real-time environmental monitoring, and in-situ chemical analysis. Beyond extending the capabilities of lithium niobate-based integrated platforms, our work represents a stride toward multifunctional on-chip sensors, ushering in a new era in integrated photonics and spectroscopic technologies.

## Methods

### Resonance evaluation

The resonance frequency of the lithium niobate multi-functional platform (LN-MFP) was characterized through electrical excitation to optimize its performance for spectroscopic applications. The experimental setup is depicted in Supplementary Fig. 1. A function generator, controlled by LabView software, produced sinusoidal voltage signals with a constant amplitude of 40 mV and frequencies ranging from 10,470 Hz to 10,500 Hz. These signals were applied to one terminal of the LN-MFP, inducing mechanical vibrations through the piezoelectric effect. The output current from the LN-MFP, which corresponds to its mechanical response, was converted into a voltage signal using a transimpedance amplifier with a 10 MΩ feedback resistor. This amplified signal was then processed by a lock-in amplifier, which was synchronized with the excitation frequency generated by the function generator. By selectively amplifying signals at the excitation frequency while suppressing noise at other frequencies, the lock-in amplifier enhanced the signal-to-noise ratio, ensuring precise detection and analysis of the LN-MFP's vibrational response.

The demodulated signals were recorded and analyzed by a computer, enabling precise characterization of the LN-MFP's resonance behavior. To accurately determine the center resonance frequency, a Lorentzian fitting was applied to the resonance curve. This precise frequency identification is essential for optimizing the device's sensitivity and selectivity, as the resonance frequency plays a critical role in amplifying mechanical vibrations induced by photoacoustic and photothermal-elastic effects.

### Experimental setup

The experimental setup for photoacoustic and light-induced thermo-elastic detection using the LN-MFP is illustrated in Supplementary Fig. 2. A function generator provided modulation signals to the laser driver, utilizing either amplitude modulation (AM) or wavelength modulation (WM) depending on the characteristics of the light source. For non-tunable light sources, such as light-emitting diodes (LEDs), AM was employed by directly applying square or sinusoidal modulation signals to modulate the light intensity. In contrast, for tunable semiconductor lasers, including laser diodes (LDs), ICLs, or QCLs, WM was implemented by superimposing a sinusoidal signal onto a triangular waveform. The slow triangular wave enabled wavelength scanning by varying the injection current, while the superimposed sinusoidal dither signal enabled high-frequency modulation for lock-in detection.

The modulated light was collimated and directed either through the LN-MFP tines or onto its surface, depending on the detection mode. In PAS detection, the collimated light beam passed through the gap between the LN-MFP tines, where it interacts with gas molecules in the chamber. This interaction led to modulated molecular absorption, generating acoustic waves that were subsequently detected. In LITES detection, the light was instead focused directly on the LN-MFP surface, where it induced thermal expansion and mechanical deformation through the thermoelastic effect.

The LN-MFP was enclosed within a gas chamber designed to enable controlled interaction with target gas molecules. The gas flow system comprised standard gas cylinders supplying calibrated analyte concentrations, mass flow controllers for precise flow regulation, pressure gauges for system stabilization, temperature controllers to ensure consistent operating temperatures, and a vacuum pump for efficient gas circulation. This setup provided precise control over gas composition and environmental conditions, ensuring optimal conditions for spectroscopic measurements.

Electrical signals generated by the LN-MFP, resulting from its mechanical response to the modulated light, were first amplified using a transimpedance preamplifier. The amplified signals were then demodulated by a lock-in amplifier synchronized with the modulation frequency, effectively enhancing detection sensitivity by filtering out noise and isolating the signal corresponding to target gas absorption. The lock-in amplifier was configured with a 1 s time constant and a 12 dB/octave low-pass filter slope, corresponding to an effective detection bandwidth of 0.25 Hz. Processed signals were recorded using data acquisition software, and gas concentrations were determined based on the signal amplitude and pre-established calibration curves.

For alignment and verification, the transmitted or reflected light beam was captured by a camera after interacting with the LN-MFP. This enabled real-time monitoring of optical alignment, ensuring optimal interaction between the light and the sensing platform.

## Data availability

The data for Figs.1–5 are provided in the source data file. The raw data are available from the corresponding author upon request. Source data are provided with this paper. Source Data file has been deposited in the Figshare [https://doi.org/10.6084/m9.figshare.30436351].

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

## Acknowledgements

This work is supported by National Natural Science Foundation of China 62375111 (H.Z.), 62005105 (H.Z.), 12174156 (W.Z.), 12174155 (J.Y.), Fundamental and Interdisciplinary Disciplines Breakthrough Plan of the Ministry of Education of China JYB2025XDXM802 (H.Z.), the Ministry of Education of China 8091B03012309 (H.Z.), Natural Science Foundation of Guangdong Province 2023B1515020027 (H.Z.), Beijing Natural

Science Foundation L254015 (H.Z.), the Science and Technology Projects of Guangzhou 2025A04J5212 (H.Z.), the Higher Education Institutions of Guangdong Province 2025ZDZX1001 (H.Z.), Guangdong Special Support Program 2024TQ08A171 (H.Z.), the Fundamental Research Funds for the Central Universities 21619402, 1161 8413, 21624113 (H.Z), State Key Laboratory of Applied Optics SKLAO-201914 (H.Z.). Special Funds for the Cultivation of Guangdong College Students' Scientific and Technological Innovation NO. pdjh2024a051 (H.Lin), NO. pdjh2024a048 (H.Lin), National Innovation and Entrepreneurship Training Program for Undergraduate 202410559001 (H.Lin), 202510559033 (H.Lin). Vincenzo Spagnolo acknowledges funding from PNRR MUR project PE0000023-NQSTI and MUR-Dipartimenti di Eccellenza 2023 – 2027 project Quantum Sensing and Modeling for One-Health (QuaSiModO). The authors thank Professor Tuan Guo for discussions and suggestions. The authors also thank Shuiliang Cao from the Analytical and Testing Center of Jinan University for SEM imaging.

## Author contributions

H.Z., A.S., P.P., and V.S. conceived the idea. H.L., H.Lu, J.Yu, and H.Z. designed the experiment. H. Lin conducted the experiment. W.Z., Y.Z., Z.J., and H.W. analyzed the data. H.L., A.S., and P.P. finished the manuscript. J.Z., X.J., V.S., and L.D. supervised the project.

## Competing interests

The authors declare no competing interests.
