## [Peer Review File · Nature Communications]

Multifunctional lithium niobate platform for photodetection and photoacoustic and thermoelastic gas sensing

Corresponding Author: Professor Lei Dong

Version 0:

Reviewer comments:

Reviewer #1

(Remarks to the Author)

This manuscript presents an integrated lithium niobate platform (LN-MFP) that combines photoacoustic spectroscopy, light-induced thermoelastic spectroscopy, and broadband photodetection into a single fork-shaped piezoelectric device. The authors demonstrate trace gas detection over a broad spectral range, and the integration of multiple sensing functionalities within a single lithium niobate structure is promising, especially in terms of compact photonic sensor development. The experimental results clearly show the device functions. However, I believe the manuscript, in its current form, does not meet the publication standards of Nature Communications due to substantial limitations in rigor, design transparency, and technical completeness.

1. The manuscript provides only schematics (Fig. 1a and Fig. 5a) without actual images of the fabricated LN-MFP device and the integrated QCL-LN system. Given that this work claims a significant advancement in integration and experimental realization, the absence limits the manuscript's transparency and reproducibility. I strongly recommend the authors provide those below.

- (1) Optical microscope and SEM images of the fabricated LN-MFP
- (2) Photographs of the packaged sensor, including the wire-bonded QCL and LN-MFP on the PCB
- (3) Actual images of the experimental setup used for sensing demonstrations

2. The manuscript introduces a fork-shaped LN structure but does not provide any systematic design optimization or performance trade-off analysis. While dimensions are reported, there is no discussion of how the geometry was selected to optimize resonance, signal strength, or multifunctional operation. In addition, I strongly recommend the authors clarify whether the geometry was optimized specifically for PAS, LITES, photodetection, or a compromise among them.

3. The MDLs are estimated using 1-sigma noise, but the authors do not specify detailed measurement conditions, such as (1) lock-in amplifier time constant, (2) detection bandwidth, and (3) integration time or sampling rate. I believe this omission limits reproducibility and meaningful comparison with other platforms. In addition, it is strongly recommended to include Allan deviation analysis to evaluate long-term stability and optimal averaging time, as is standard practice in gas sensing.

4. The description of light coupling between the QCL chip and the LN-MFP is vague. The ~100 μm free-space coupling distance is mentioned without supporting figures, optics specification, beam alignment tolerance, or coupling efficiency estimation. I believe that this is critical given the compact packaging and the divergence of QCL beams in the mid-IR. I recommend the authors (1) provide a photo and diagram of the actual optical alignment, (2) specify whether any collimating optics were used and, (3) discuss the stability of this coupling under practical conditions.

5. Please include a calibration curve or concentration-response plots for the target gas species in the measurements with the LN-MFP device.

6. Please provide the temporal response time of each sensing mode.

7. The manuscript lacks sufficient detail regarding the fabrication process of the LN-MFP device. To improve reproducibility and technical clarity, please provide a more specific and comprehensive description of the process flow, including details on substrate preparation, lithographic patterning, etching methods (e.g., ICP parameters, mask materials), metallization, and

any relevant bonding or release techniques.

8. A summary/benchmark table comparing the LN-MFP sensor's performance metrics (e.g., MDL, responsivity, bandwidth, response time) with existing PAS, LITES, and pyroelectric sensors would enhance the impact of the work.

Reviewer #2

(Remarks to the Author)

The manuscript describes a lithium niobate multifunctional integrated platform (LN-MFP) for spectroscopic gas sensing. Specifically, the authors present a systematic study of a lithium niobate tuning fork for photoacoustic, thermoelastic/photodetection-based sensing. It is of interest to see the direct bonding of the QCL chip to the LN-MFP, showing potential for integrated spectroscopic applications. This work will attract researchers from different areas such as spectroscopy, sensing, integrated photonics, etc. The following issues should be addressed before this work can be considered for publication.

1. One key application shown in this study is photoacoustic spectroscopy. The authors show a good record of adopting tuning forks (especially quartz tuning fork, QTF) in gas sensing with demonstrated high sensitivity across a broad spectral range. However, in the manuscript, the authors seem to only report the PAS gas sensing results without providing sufficient discussion. A detailed comparison with state-of-the-art tuning fork-based PAS gas sensing systems is needed to highlight the key advantages and advancements offered by the designed LN tuning fork.

2. In addition, the following information should be added to strengthen the technical content:

- (1) Experimental setup for each light source is suggested to be documented in the supplement.
- (2) Better to include N₂ background signals in Fig. 2 for direct comparison.

3. Regarding the demonstration of thermoelastic detection (LITES), the reviewer notices tuning forks have achieved high sensitivity in the visible, near-IR, and mid-IR. The manuscript should compare with state-of-the-art LITES results and clarify how the LN tuning fork advances beyond these results.

4. Following the previous question, some specific concerns:

- (1) The claim that "resonant amplification significantly boosts sensitivity" is confusing, as the LITES performance is notably worse than PAS. It is noticed that the author used a 2.5 cm optical length, but the results are still unsatisfied to support your claims. Further discussion is needed to clarify this discrepancy.
- (2) Clarify how dual PAS/LITES functionality enables comprehensive analysis. It is not straightforward for readers to understand how combining them together can leverage the sensing power of LN-MFP.
- (3) Clarify the relationship between spectral selectivity and dual functionality, as selectivity is primarily determined by ro-vibrational fingerprints.

5. It seems there is an overlap between the photodetection functionality with the thermoelastic sensing mechanism. Is it more reasonable to move this section before LITES as it relates to the fundamental mechanism? In addition, the modulated photodetection using tuning forks has been previously demonstrated, so it is more interesting to make a comparison with state-of-art photodetection using tuning forks, highlighting key pros and cons. Also, it is better to make a fair comparison of noise-equivalent power (NEP) with commercial photodetectors, which is more important in photodetection.

6. LN is an important photonic platform used in many areas as described in the first paragraph of Introduction. However, the authors missed mentioning the spectroscopic sensing applications of using LN. May be not too many in this topic, but at least there are a couple of pioneering work recently reported on fabricating LN rib waveguide and using it for on-chip photothermal gas sensing. As spectroscopic sensing the key topic of this paper, the authors should make a more complete literature to strengthen this part.

Additional concerns:

- (1) Why is the tuning fork placed horizontally rather than vertically?
- (2) It is more interesting to include a photo/microscope image of the on-chip LN-MFP sensor in Figure 5.
- (3) The results show the signal of 8000 ppm CO. Please discuss limitations and potential solutions.

7. Regarding the design and fabrication of the LN-MFP, a discussion and comparison with existing LN, quartz, silicon, and polymer tuning forks should be added. The key parameters (resonant frequency, quality factor) should be moved to the main text, with detailed simulation and characterization data properly referenced in supplementary materials. The key parameters like resonant frequency, quality factor, and dimensions will be useful for readers to understand the advancements of the proposed device.

Reviewer #3

(Remarks to the Author)

The paper presents a new device made from lithium niobate (LiNbO₃) that can do three things: detect sound made by light (photoacoustic), sense heat changes (thermoelastic), and respond to light across a wide range from visible to long-wave infrared. The experiments show it could be useful for small integrated sensors. The authors used the special electric and optical properties of LiNbO₃ to build a small sensor that does multiple things, which is both creative and technically important. I think this paper could be published if some issues are fixed. Here are my concerns:

-The paper says they added an acoustic resonator around the LN-MFP to make the signal stronger. According to the

Supporting Information, the signal became 79 times bigger. But how big is the resonator compared to the tuning fork? How are they placed relative to each other? Did they try to match the size of the resonator to the frequency of the tuning fork? Also, did they test or simulate how well the resonator and tuning fork work together?

-In the thermoelastic detection part, the laser was pointed at the side of the tuning fork prong instead of the front. Why was that done? Does lighting from the side help create more thermal stress or better vibration? Were any comparisons done between side and front illumination to see which works better?

-The paper shows simulations saying that the shape and position of the electrodes affect how the device vibrates. But there's no experiment to prove this. Have the authors made real devices with different electrode shapes and tested them, like checking their frequency response or how they vibrate?

-The authors used a Y-cut 128° LiNbO₃ crystal but didn't explain why. Since LiNbO₃ has different properties depending on how it's cut (like Z-cut, X-cut, etc.), maybe this cut gives better piezoelectric effect or mechanical quality. Could the authors say what makes this cut better for their device? Did they compare it with other cuts in simulations or tests?

-The paper mentions using a lock-in amplifier to get the 2f signal, but doesn't say how it was set up. Important settings like time constant, filter slope, reference phase, and bandwidth can affect the signal quality. Could the authors give more details about these settings?

-In many sensing methods like QEPAS and LITES, quartz is usually used because it's stable and has good resonance. Why did the authors choose LiNbO₃ instead? Is it because it can do more than one thing, has better sensitivity, or can be used with light on a chip? A comparison with quartz would help explain this choice.

-The paper combines photoacoustic, thermoelastic, and photodetection into one device, but it's not clear why all three are needed together. What advantages does this combination give? It would help to have a comparison with using them separately.

-For the photodetection tests, the authors changed the light power to see how the device responds. But since LiNbO₃ absorbs different wavelengths differently, how do they know the change in signal is only because of power and not wavelength? Were the measurements done at fixed wavelengths? Or were power and wavelength controlled separately?

-In the conclusion, the authors talk about making a small photonic chip from this platform. But how will they turn this lab-scale device into something small enough for a chip? There are challenges like tiny fabrication, connecting parts on a wafer, and packaging. Also, how will they add light sources directly onto the same chip?

-The paper introduces a new multi-purpose LiNbO₃ platform, but it doesn't compare it clearly with existing technologies like QEPAS or LITES. Could the authors give a table or explanation comparing key performance numbers like detection limit, speed, and noise equivalent power?

Version 1:

Reviewer comments:

Reviewer #1

(Remarks to the Author)

The authors have addressed my previous comments well overall, and the quality of the revised manuscript has improved. However, I still have several remaining concerns.

1. While the multifunctional operation of the LN-MFP platform is interesting, the level of novelty remains limited. The individual sensing mechanisms (PAS, LITES, and thermoelastic photodetection) have been demonstrated previously using QTF-based and MEMS-based systems, and the performance improvements presented here appear incremental rather than introducing a fundamentally new advancement. The manuscript should clearly describe the specific scientific or technological contribution that is unique to this LN-based approach, and how it advances the current state of the art. I believe that strengthening this aspect is very important for meeting the expectations of Nature Communications.

Additionally, the term "on-chip integration", especially in the abstract, may lead readers the impression of monolithic or wafer-scale integration. In this work, however, the demonstrated QCL-LN configuration is based on PCB-level co-packaging rather than true chip-scale integration. I recommend clarifying this distinction in both the abstract and the main text to avoid overstating the integration level. In addition, it would be helpful for the authors to more explicitly describe what the key novelty and strengths of this work are, although the integration remains at the PCB level. Providing a clear justification of the scientific value and technological impact of this work, despite not achieving on-chip integration, would strengthen the manuscript's positioning.

2. Although the authors provide additional details regarding the free-space alignment between the QCL and the LN-MFP, the optical coupling efficiency is still not reported. Since key performance metrics such as NNEA and MDL depend on the optical power actually delivered to the sensing region, the absence of a coupling-efficiency estimate introduces uncertainty

in the quantitative interpretation of the results. Please clarify how the NNEA values were calculated rigorously.

Reviewer #2

(Remarks to the Author)

The authors have made significant revision in the manuscript and well answered most of my questions. Here I have only a few minor comments for authors' consideration and clarification.

- (1) I appreciate the authors added N₂ background signals in Fig. 2 for direct comparison, but there is significant non-zero signal in (b) and (c). Please explain and mention whether it is an issue for applications.
- (2) The authors made detailed response to the previous question Q3 regarding comparison to LITES. I notice this part has been added to supplemental, but nothing is mentioned in the main text. Please inform readers this part is discussed in the supplemental information.
- (3) In the previous question Q5, the Table 3s compares pyroelectric detectors. Is it possible to include photovoltaic detector in the table as well, considering its wide use in spectroscopy.

Reviewer #3

(Remarks to the Author)

The authors have carefully revised the manuscript according to my comments. I am content with the revisions and suggesting acceptance now.

Response to reviewers' comments

Dear Reviewers:

Please find attached our point-by-point responses to all comments from the three reviewers. We have revised the manuscript accordingly.

Please Note: reviewers' comments are in black; our comments are in *red italic*; the original paper text is in red and revised text is in blue.

Reviewer #1:

This manuscript presents an integrated lithium niobate platform (LN-MFP) that combines photoacoustic spectroscopy, light-induced thermoelastic spectroscopy, and broadband photodetection into a single fork-shaped piezoelectric device. The authors demonstrate trace gas detection over a broad spectral range, and the integration of multiple sensing functionalities within a single lithium niobate structure is promising, especially in terms of compact photonic sensor development. The experimental results clearly show the device functions. However, I believe the manuscript, in its current form, does not meet the publication standards of Nature Communications due to substantial limitations in rigor, design transparency, and technical completeness.

1. The manuscript provides only schematics (Fig. 1a and Fig. 5a) without actual images of the fabricated LN-MFP device and the integrated QCL-LN system. Given that this work claims a significant advancement in integration and experimental realization, the absence limits the manuscript's transparency and reproducibility. I strongly recommend the authors provide those below.

(1) Optical microscope and SEM images of the fabricated LN-MFP

We sincerely thank the Reviewer for this constructive suggestion. Optical-microscope and SEM images of the fabricated LN-MFP have now been added to Fig. 1 (panels d–f) in main article page 6.

“Fig. 1 | Lithium Niobate Multi-Functional Platform. a Design of the LN-MFP. **b** Simulated surface charge distribution of LN-MFP during vibration. The red regions correspond to positive charge accumulations, blue regions to negative ones. **c** Manufacturing process of LN-MFP; **d** Optical microscope image of the fabricated LN-MFP; **e** SEM image of the edge of the LN-MFP tine. **f** SEM image of the fillet of LN-MFP base.”

Related descriptions are added in main article page 5 line 24:

“Figure 1d shows an optical microscope image of the fabricated LN-MFP, while Fig. 1e and Fig. 1f present SEM images of the LN-MFP tine edge and the base fillet, respectively.”

(2) Photographs of the packaged sensor, including the wire-bonded QCL and LN-MFP on the PCB

Thanks for the comment. The photograph of the packaged sensor was added to the main article page 18, Fig.5a.

The wire-bonded QCL chip and the LN-MFP are co-mounted on a common silicon carrier and installed on a 3.5 cm × 4.5 cm PCB that integrates the readout electronics. The QCL emission facet was aligned to the LN-MFP prong gap under an optical microscope. The free-space separation between the QCL facet and the prong edge is ~100 μm.

Fig. 5 | On chip LN-MFP sensor. a Photograph of the packaged on-chip sensor.

Related descriptions are added in main article page 17 Line 23. We replaced:

“As shown in Fig. 5(a), we realized a custom-designed 3.5 cm×4.5 cm printed circuit board (PCB) with integrated transimpedance amplifier function to pre-process the charge output of LN-MFP and a QCL chip.”

with

“As displayed in the photograph shown in Fig. 5a, the wire-bonded QCL chip and LN-MFP are co-mounted on a common silicon carrier and installed on a custom 3.5 cm × 4.5 cm PCB that integrates the drive and readout electronics, including a transimpedance amplifier to pre-condition the LN-MFP charge output.”

(3) Actual image of the experimental setup used for sensing demonstrations

A photo of the experimental setup was added in Supplementary Section 10, page 17 Fig. 7s.

Fig. 7s | Photo of the employed experimental setup. PC: personal computer, DM: detection module.

Related descriptions are added to the Supplementary page 16 line 8:

“Figure 7s shows a photo of the experimental setup used for the LN-MFP measurements. A function generator (Tektronix AFG3102) supplied the modulation waveform to the light sources. The sources, as listed in the Table 5s, were driven and temperature-stabilized by a current/TEC controller (Thorlabs ITC4005QCL). The emitted beam was routed through the detection module. A camera (Dataray WinCamD-IR-BB) was employed for beam profiling and

alignment. The electrical output from the detection module was demodulated by a lock-in amplifier (SRS SR830). A personal computer was used for instrument control and data acquisition.”

Table 5s. The wavelength and model of seven light sources.

Laser source	Model	Wavelength (nm)	Power (mW)	Modulation method	IC (mA)	Temperature (°C)
VIS LED	NICHIA 450	450	1200	AM	600	/
NIR LD	NTT Electronics Corporation	1390	19	FM	84	28
NIR LD	NTT Electronics Corporation	1536	3700*	FM	90	25
NIR LD	NTT Electronics Corporation	2004	10.6	FM	120	18
MWIR ICL	NANOPLUS	3370	6.5	FM	86	17
QCL Chip	Chinese Academy of Sciences	4590	14	FM	270	45
LWIR QCL	HEATHYPHOTO N	9770	82.3	FM	420	20

IC: injection current; AM: amplitude modulation; FM: frequency modulation; *Amplified by EDFA

2. The manuscript introduces a fork-shaped LN structure but does not provide any systematic design optimization or performance trade-off analysis. While dimensions are reported, there is no discussion of how the geometry was selected to optimize resonance, signal strength, or multifunctional operation. In addition, I strongly recommend the authors clarify whether the geometry was optimized specifically for PAS, LITES, photodetection, or a compromise among

them.

We thank the reviewer for this important comment. Below we detail the design objectives and the technical rationale behind the chosen fork geometry. We describe the systematic study performed to identify the final dimensions as a practical compromise among photoacoustic (PAS), light-induced thermoelastic (LITES) and photodetection performance. The following content has already been added to the Supplementary page 3 section 2.

“The resonance frequency and quality factor (Q) of the LN-MFP are governed by both material properties and geometric configuration. In this work, the geometry was carefully optimized to balance the competing requirements of its multifunctional operation, including PAS, LITES, and photodetection. To guide this optimization, we present a developed systematic design procedure detailing how the geometric parameters of the LN-MFP were selected and refined.

(1) Optimization for Resonance

The geometry and resonance of the LN-MFP were optimized by varying two key geometric parameters: the tine length l and width w , while keeping all other parameters fixed (see Fig. 1s a for definitions). A parametric finite-element sweep of l and w revealed that the fundamental resonance f_0 decreases monotonically with increasing l and with decreasing w (Fig. 1s b). In the simulated design space f_0 ranged from ~ 34.5 kHz ($l = 6$ mm, $w = 2$ mm) to ~ 2.8 kHz ($l = 18$ mm, $w = 1$ mm).

For the LN-MFP, excessively high resonant frequency f_0 yields shorter vibration period, limiting the time available for the device to accumulate acoustic energy. Since the light-source modulation must match the LN-MFP resonance frequency, high f_0 values require faster modulation rates, which reduce the efficiency of photoacoustic generation for molecules with long vibrational-to-translational (V–T) relaxation times. For example, CO₂ exhibits V–T relaxation times on the order of 100 μ s (~ 10 kHz), and modulation beyond this frequency significantly diminishes photoacoustic generation efficiency. By contrast, very low f_0 increases susceptibility to low-frequency environmental disturbances such as mechanical vibration, air currents and $1/f$ (pink) noise.

Balancing these effects, an optimal frequency $f_0 \sim 10.5$ kHz was selected, corresponding

to tine dimensions of $l=11.5$ mm, $w=1.7$ mm.

Fig. 1s | **a** Schematic of the LN-MFP showing the defined geometric parameters: tine length l , base width W , tine width w , inter-tine gap g , thickness t , and fillet radius r . **b** Finite-element simulation of the fundamental resonant frequency f_0 as a function of tine length l and width w . The results show that f_0 decreases markedly with increasing l and with decreasing w ; spanning a frequency range from ~ 34.5 kHz to ~ 2.8 kHz across the design space.

(2) Optimization for PAS

The LN-MFP geometry was optimized for photoacoustic spectroscopy (PAS) through coupled acoustic–structural finite-element simulations. In the acoustic excitation model, a cylindrical pressure source was positioned within the tine gap to reproduce the spatial distribution of the photoacoustic pressure wave and to induce symmetric tine oscillation (see Fig. 2s a). For each geometric configuration the total piezoelectric charge generated by the device was calculated.

The resulting contour map of normalized piezoelectric charge exhibits a clear and monotonic dependence on tine dimensions (Fig. 2s b). The PAS signal strength increases with greater tine length l and with smaller tine width w . Physically, longer and more slender tines exhibit larger tip displacements and stronger strain localization near the tine base under acoustic excitation, hereby generating a higher piezoelectric charge output and enhancing signal amplitude.

Fig. 2s | **a** Simulated deformation and strain distribution of the LN-MFP under PAS; **b** Normalized piezoelectric charge as a function of tine length l and tine width w .

(3) Optimization for LITES and photodetection

Because both LITES and direct photodetection originate from light-induced heating, they were analyzed within a unified thermoelastic framework. In this model, a localized heat source was applied at the LN-MFP base to simulate the photothermal excitation induced by incident light (see Fig. 3s a). For each geometric configuration, the total piezoelectric charge collected by the electrode was computed. The resulting contour map of normalized piezoelectric charge in (Fig. 3s b) reveals a systematic increase with the tine width w and only a weak dependence on tine length l . This trend arises from several factors. First, wider tines provide a larger heated volume and absorption area for a given illumination spot, leading to greater overall thermal expansion and a larger bending moment. Second, increasing w enlarges the electromechanically active volume and electrode-covered area, enhancing conversion of localized strain into piezoelectric charge. Third, because the thermoelastic excitation is concentrated near the base, variations in tine length l exert minimal influence on the thermal-to-mechanical coupling in this region.

Fig. 3s | **a** Simulated deformation and strain distribution of the LN-MFP under LITES and photodetection excitation; **b** Contour map of normalized piezoelectric charge as a function of tine length l and tine width w .

(4) Other geometry parameters

-Tine gap g

To accommodate diverse optical sources, from visible LEDs to mid-IR QCLs, and to balance optical and acoustic requirements, the tine gap was fixed at $g=1.0$ mm. The gap size is governed by two competing effects. For LITES and photodetection, light is incident on the outer tine surfaces, whereas in PAS operation, the excitation beam must traverse the gap without contacting the electrodes. If the gap is too narrow, the laser spot overlaps the metalized tines, introducing significant photothermal background noise. Conversely, if the gap is too wide, the acoustic pressure field weakens, reducing acoustic-to-mechanical coupling efficiency. Therefore, a 1.0 mm gap represents a practical compromise between these competing optical and acoustic constraints.

- Device thickness t

Commercial lithium-niobate wafers are typically available in thicknesses of ~ 500 μm . Within the explored design-space, variation in device thickness produced only minor effects on the fundamental resonance compared to the dominant influence of tine length and width. Its impact on electromechanical and thermoelastic responses was similarly limited. Considering

factors such as wafer availability, fabrication precision, mechanical robustness and process repeatability, a thickness of $t=500\ \mu\text{m}$ was selected for the fabricated LN-MFP devices.

(5) Conclusion

In summary, a comprehensive finite-element parameter sweep combined with multiphysics simulation was conducted to optimize the LN-MFP geometry across three key performance dimensions: resonant behavior, photoacoustic (PAS) response, and thermoelastic/photodetection response. The inherently opposing design requirements of PAS and LITES/photodetection necessitated a balanced configuration. The finalized structure, with $l=11.5\ \text{mm}$, $w=1.7\ \text{mm}$, $g=1.0\ \text{mm}$, and $t=500\ \mu\text{m}$, exhibits a fundamental resonant frequency near 10.5 kHz. This optimized design achieves high PAS sensitivity alongside strong LITES/photodetection responsivity, ensuring multifunctional operation with excellent mechanical stability, fabrication compatibility and system integration compatibility.”

3. The MDLs are estimated using 1-sigma noise, but the authors do not specify detailed measurement conditions, such as (1) lock-in amplifier time constant, (2) detection bandwidth, and (3) integration time or sampling rate. I believe this omission limits reproducibility and meaningful comparison with other platforms. In addition, it is strongly recommended to include Allan deviation analysis to evaluate long-term stability and optimal averaging time, as is standard practice in gas sensing.

Many thanks for the comment. According to the reviewer’s suggestion, we added the requested measurement conditions and Allan deviation analysis to revised manuscript.

In main article page 25 line 4, we added:

“The lock-in amplifier was configured with a 1 s time constant and a 12 dB/octave low-pass filter slope, corresponding to an effective detection bandwidth of 0.25 Hz.”

Regarding the Allan deviation analysis, for PAS, we added the following sentences in main article page 8 line 18:

“To assess the long-term stability of the LN-MFP PAS sensor, we performed noise characterization and Allan-deviation analysis (see Fig. 2). The 1σ noise levels were measured under a continuous flow of 200 sccm pure N_2 . Figures 2b–g show the corresponding noise traces

and Allan-deviation curves for the LN-MFP operated with a 450 nm LED, 1392 nm laser diode, 1536 nm laser diode, 2004 nm laser diode, 3370 nm ICL, and 9770 nm QCL. From the Allan deviation analysis, the 1σ MDLs, and optimal averaging times were extracted: the 450 nm (NO_2) channel achieves an MDL of ~ 2 ppb at an averaging time of 1,000 s; the 1,392 nm (H_2O), ~ 25 ppb at 6,300 s; the 1,536 nm (C_2H_2), ~ 3.5 ppb at 1000 s; the 2,004 nm (CO_2), ~ 350 ppb at 1,900 s; the 3,370 nm (CH_4), ~ 2.5 ppb at 1,200 s; and the 9,770 nm (NH_3), ~ 350 ppt at 1,100 s.

Fig. 2 | LN-MFP for photoacoustic detection. **a** Spectral coverage with excitation sources from visible to long-wave infrared and the targeted analytes. **b-g** present representative measurements for each target gas: the main plot in each panel shows a typical $2f$ signal acquired with the indicated light source and sample concentration; the left inset shows the concentration-calibration (linear fit); the top-right inset shows a 1σ noise level obtained with pure N_2 ; and the bottom-right inset shows the Allan-deviation curves used to extract the 1σ minimum detection limit (MDL) and optimal averaging times. Extracted MDLs (1σ) and averaging times are: **b**

MDL \sim 2 ppb at 1,000 s; **c** MDL \sim 25 ppb at 6,300 s; **d** MDL \sim 3.5 ppb at 1,000 s; **e** MDL \sim 350 ppb at \sim 1,900 s; **f** MDL \sim 2.5 ppb at 1,200 s; **g** MDL \sim 350 ppt at 1,100 s. VIS LED: visible light emitting diode; NIR LD: near-infrared laser diode; MWIR ICL: mid-wave interband cascade laser; LWIR QCL: long-wave quantum-cascade laser.

And also, for LITES, in main article page 16 line 10:

“To assess the long-term stability of the LN-MFP LITES sensor, we performed measurements identical to those used for the PAS characterization and computed Allan-deviation curves for each detection channel. From the Allan deviation analysis, the 1σ MDLs, and the corresponding optimal averaging times were extracted. The 450 nm (NO_2) channel achieves an MDL of \sim 1 ppm at an optimal averaging time of 450 s; the 1,392 nm (H_2O) channel, \sim 2.5 ppm at 1,600s; the 1,536 nm (C_2H_2) channel, \sim 450 ppb at 1500 s; the 2,004 nm (CO_2) channel, \sim 35 ppm at 1,200 s; the 3,370 nm (CH_4) channel, \sim 900 ppb at 1,900 s; and the 9,770 nm (NH_3) channel \sim 1.2 ppm at 1,200 s.”

Fig. 4 | LN-MFP for thermoelastic detection. **a** Spectral coverage of excitation sources from visible to long-wave infrared, together with the corresponding targeted analytes. **b-g** Representative measurements for each target gas species: the main plots shows typical $2f$ signal acquired with the indicated light source and gas concentration. The left inset displays the concentration-calibration (linear fit); the top-right insets show the 1σ noise level measured for pure N_2 ; and the bottom-right insets present the Allan-deviation curves used to extract the 1σ MDL and optimal averaging times. Extracted MDLs (1σ) and optimal averaging times are: **b** MDL ~ 1 ppm at 450 s; **c** MDL ~ 2.5 ppm at 1,600 s; **d** MDL ~ 450 ppb at 1500 s; **e** MDL ~ 35 ppm at 1,200 s; **f** MDL ~ 900 ppb at 1,900 s; **g** MDL ~ 1.2 ppm at 1,200 s.

4. The description of light coupling between the QCL chip and the LN-MFP is vague. The ~ 100 μm free-space coupling distance is mentioned without supporting figures, optics specification, beam alignment tolerance, or coupling efficiency estimation. I believe that this is critical given the compact packaging and the divergence of QCL beams in the mid-IR. I recommend the authors (1) provide a photo and diagram of the actual optical alignment, (2) specify whether any collimating optics were used and, (3) discuss the stability of this coupling under practical conditions.

We thank the reviewer for this valuable comment.

(1) provide a photo and diagram of the actual optical alignment

We have replaced the original Fig. 5 with an expanded Fig. 5 in main article page 18 to show the photo and diagram.

Fig. 5 | On chip LN-MFP sensor. **a** Photograph of the packaged on-chip sensor. **b** Optical-microscope image of the coupling region showing the QCL emission facet aligned with the LN-MFP prong gap. **c** Schematic diagram of the free-space optical coupling. **d** $2f$ photoacoustic spectrum of 8,000 ppm CO in N₂ measured using the integrated LN-MFP–QCL assembly.

The expanded Fig.5 includes: a a high-resolution photograph of the packaged on-chip assembly showing the wire-bonded QCL chip and the LN-MFP on the silicon carrier; b an optical-microscope close-up of the coupling region showing the QCL facet and the tine gap; and c a schematic that reproduces the measured geometry and beam path.

(2) specify whether any collimating optics were used

In main article, page 18 line 5, we add the following sentence:

“As shown in Fig. 5b and 5c, no collimating optics were used in the system.”

No external collimating or focusing optics were employed in the on-chip coupling experiments described in this work. The QCL was operated as a bare die, emitting freely across the gap toward the LN-MFP. Prior to alignment, the QCL emission pattern was characterized using a mid-IR camera to verify beam divergence and ensure proper coupling geometry.

(3) discuss the stability of this coupling under practical conditions

As shown in Fig. 5b–c, the QCL chip was aligned directly to the LN-MFP tine gap without the use of collimating optics. The QCL far-field, measured with a mid-IR camera, exhibited a divergence of $\sim 70^\circ$. The tine gap was measured under an optical microscope as ≈ 1.0 mm, and the separation between the QCL and LN-MFP was ~ 100 μm . The QCL chip was soldered on a silicon carrier, and after fine alignment under the microscope, the LN-MFP was fixed in position using conductive silver paste. This simple free-space coupling configuration enables proof-of-concept PAS measurements (see Fig. 5d). Because no precision collimation optics were employed, the coupling tolerances are relatively relaxed. Only a portion of the emitted light needs to overlap the tine gap to generate a detectable signal, which simplifies on-chip alignment and ensuring mechanical stability. However, the absence of beam shaping elements also allows partial stray illumination of the tine faces, introducing photothermal background, reducing optical coupling efficiency and potentially increasing noise that limits long-term stability.

To further improve coupling efficiency and robustness, future designs will explore: (i) the integration of a micro-collimator or compact mid-IR lens between the QCL facet and the tine gap to reduce divergence and enhance beam overlap with the gap; (ii) incorporation of a molded microlens or etched micro-optic on the silicon carrier; (iii) monolithic on-chip coupling strategies such as integrated waveguide or butt-coupling to eliminate free-space tolerances; and (iv) hermetic, thermally-matched packaging with mechanical stabilization to minimize drift during thermal cycling.

To make it clear to the readers, the following sentences have been added to the Supplementary page 13 Section 8.:

“(1) Optical-alignment and stability.

The QCL chip was aligned directly to the LN-MFP tine gap without external collimating optics. The QCL far-field characterized using a mid-IR camera, exhibited a divergence of $\sim 70^\circ$. The tine gap, under an optical microscope was ~ 1.0 mm and the separation between QCL facet and LN-MFP was ~ 100 μm . The QCL chip was soldered to a silicon carrier, and after fine optical alignment under the microscope, the LN-MFP was secured in position with conductive silver

paste. This free-space coupling configuration enables robust proof-of-concept PAS measurements with relaxed lateral alignment tolerances.

(2) Limitations and mitigation.

In the absence of beam-shaping optics, part of the emitted power falls outside the LN-MFP tine gap, leading to stray illumination of the tine surfaces. This produces photothermal background signals, reduces optical coupling efficiency, and can ultimately limit long-term stability and the achievable minimum detection limit (MDL). To enhance coupling efficiency and robustness in future iterations, several strategies will be explored: (i) integrating a micro-collimator or compact mid-IR lens between the QCL facet and the tine gap to reduce divergence and improve overlap; (ii) incorporating molded or etched microlenses on the silicon carrier to shape the beam at the chip level; (iii) implementing monolithic on-chip coupling approaches, such as integrated waveguide or butt-coupling to eliminate free-space tolerances and further stabilize the optical alignment.”

5. Please include a calibration curve or concentration-response plots for the target gas species in the measurements with the LN-MFP device.

We thank the reviewer for this important suggestion. We have added the calibration curves to Fig.2 and Fig.4 in the main article page 9 and page 14.

Fig. 2 | LN-MFP for photoacoustic detection. **a** Spectral coverage with excitation sources from visible to long-wave infrared and the targeted analytes. **b-g** present representative measurements for each target gas: the main plot in each panel shows a typical 2f signal acquired with the indicated light source and sample concentration; the left inset shows the concentration-calibration (linear fit); the top-right inset shows a 1σ noise level obtained with pure N₂; and the bottom-right inset shows the Allan-deviation curves used to extract the 1σ minimum detection limit (MDL) and optimal averaging times. Extracted MDLs (1σ) and averaging times are: **b** MDL ~2 ppb at 1,000 s; **c** MDL ~25 ppb at 6,300 s; **d** MDL ~3.5 ppb at 1,000 s; **e** MDL ~350 ppb at ~1,900 s; **f** MDL ~2.5 ppb at 1,200 s; **g** MDL ~350 ppt at 1,100 s. VIS LED: visible light emitting diode; NIR LD: near-infrared laser diode; MWIR ICL: mid-wave interband cascade

laser; LWIR QCL: long-wave quantum-cascade laser.

Related description in main article page 8 line 16:

“Responses to varying target-gas concentrations were measured and the corresponding calibration curves are plotted in Fig. 2. As shown in Fig. 2, all channels exhibit excellent linearity, with linear-fit coefficients of $R^2 > 0.99$.”

Fig. 4 | LN-MFP for thermoelastic detection. **a** Spectral coverage of excitation sources from visible to long-wave infrared, together with the corresponding targeted analytes. **b-g** Representative measurements for each target gas species: the main plots shows typical $2f$ signal acquired with the indicated light source and gas concentration. The left inset displays the concentration-calibration (linear fit); the top-right insets show the 1σ noise level measured for pure N_2 ; and the bottom-right insets present the Allan-deviation curves used to extract the 1σ MDL and optimal averaging times. Extracted MDLs (1σ) and optimal averaging times are: **b** MDL ~ 1 ppm at 450 s; **c** MDL ~ 2.5 ppm at $1,600$ s; **d** MDL ~ 450 ppb at 1500 s; **e** MDL ~ 35 ppm at $1,200$ s; **f** MDL ~ 900 ppb at $1,900$ s; **g** MDL ~ 1.2 ppm at $1,200$ s.

Related description in main article page 16 line 8:

“Similar to the PAS, calibration measurements for the LITES channels are summarized in Fig.

4. All channels exhibit excellent linearity with linear-fit $R^2 > 0.99$.”

6. Please provide the temporal response time of each sensing mode.

Thanks for the comment. In this work, the lock-in amplifier and data acquisition module were configured to output one data point per second, corresponding to a theoretical minimum response time of 1 s.

It is noteworthy that the theoretical temporal response times of PAS, LITES, and photodetection in this work are significantly shorter than 1 s. For optical detection techniques based on resonance principles, the absorbed laser power is accumulated within the resonator over Q oscillation periods, where Q denotes the quality factor of the resonator [1]. The time required for the signal amplitude to rise from zero to a steady-state level, the response time, can be estimated by the theoretical model $\tau = Q / f_0$ [2]. For the LN-MFP, with a resonance frequency $f_0 = 10,485$ Hz and a Q factor of 1,621, the corresponding theoretical response time is approximately 150 ms. However, in this work, to guarantee stable operation of the LN-MFP system and to average the signal fluctuations, the output rate was intentionally limited to one data point per second. Therefore, the effective temporal response time of the entire LN-MFP sensing system is approximately 1 s.

In the revised manuscript, we added the following sentence in the main article page 25 line 4 to provide the response time:

“The lock-in amplifier was configured with a 1 s time constant and a 12 dB/octave low-pass filter slope, corresponding to an effective detection bandwidth of 0.25 Hz.”

Reference

[1] Kosterev A A, Bakhirkin Y A, Curl R F, et al. Quartz-enhanced photoacoustic spectroscopy[J]. Optics letters, 2002, 27(21): 1902-1904.

[2] Zhang H, Jin W, Hu M, et al. Investigation and optimization of a line-locked quartz enhanced spectrophone for rapid carbon dioxide measurement[J]. Sensors, 2021, 21(15): 5225.

7. The manuscript lacks sufficient detail regarding the fabrication process of the LN-MFP device. To improve reproducibility and technical clarity, please provide a more specific and comprehensive description of the process flow, including details on substrate preparation, lithographic patterning, etching methods (e.g., ICP parameters, mask materials), metallization, and any relevant bonding or release techniques.

Thanks for the comment. According to the reviewer's suggestion, we provided a more specific and comprehensive description of the process flow in the revised manuscript and the Supplementary.

In the revised main article page 5 line 19, we add:

“A more specific and comprehensive description of the process flow was provided in the Supplementary Section 3.”

In the Supplementary page 7 line 15 Section 3, we add the following text:

“The LN-MFP device was fabricated on a double-side-polished Y+128°-cut LiNbO₃ wafer with a thickness of 500 μm. The process began with wafer slicing, edge grinding, and double-side polishing to ensure surface flatness and uniform thickness followed by ultrasonic cleaning to remove microscopic contaminants.

A Cr/Au (20/200 nm) bilayer was then deposited to form the electrode layer using an electron-beam evaporation system (model DE400, Texas Instruments, USA) under a base vacuum of 3×10^{-6} Torr. The Cr adhesion layer was evaporated at 30 W with a rate of 0.5 Å/s for 400 s, followed by deposition of the Au layer at 40 W, 0.8 Å/s for 2,500 s ensuring uniform film adhesion and high surface conductivity.

Subsequently, 355 nm UV photolithography was employed to define the electrode geometry directly on the wafer surface. Both electrode patterning and device dicing were performed using a UV laser (5 W, 20 kHz repetition rate, 30 μs pulse width), achieving maskless fabrication with ±10 μm precision. During electrode definition, each exposure lasted 10 s, sufficient to pattern the Cr/Au electrodes. For dicing, the same laser system operated for over 1200 s, fully cutting through the 500 μm-thick LN substrate.

Following laser processing, an “optical-grade dicing” and polishing step was implemented

to refine structural precision. The LN-MFPs were mounted on a UV-sensitive adhesive film attached to a stainless-steel polishing frame. A high-precision optical dicing system equipped with a 56 mm-diameter, 150 μm -thick resin-bonded diamond wheel was used to polish the sidewalls at a rotation speed of 10,000 rpm and a feed rate of 0.2 mm/s, with deionized-water circulation maintaining a stable temperature of 18 $^{\circ}\text{C}$.

Finally, the fabricated LN-MFP was integrated onto the chip via gold wire bonding technology. The bonding process used controlled heat, pressure, and ultrasonic energy to connect the surface electrodes of the LN-MFP to the silicon carrier. The bonded gold wires were 20 μm in diameter and the backside electrode was attached with conductive silver paste to ensure reliable grounding and enhanced electrical contact.”

8. A summary/benchmark table comparing the LN-MFP sensor's performance metrics (e.g., MDL, responsivity, bandwidth, response time) with existing PAS, LITES, and pyroelectric sensors would enhance the impact of the work.

Many thanks for the comment. Three detailed tables about PAS, LITES, and pyroelectric sensors were added to the Supplementary page 13 section 9:

“In photoacoustic spectroscopy, the key parameters that govern detection performance are the laser wavelength, optical power, and integration time. These parameters together with the minimum detection level (MDL), relative MDL (accounting for the proportionality of the photoacoustic signal to laser power), and normalized noise equivalent absorption (NNEA) coefficient, are summarized in Table 2s. Notably, the LN-MFP sensor is chip-scale offering inherent advantages in compactness, robustness, and ease of system integration.

Table 2s PAS detection of trace gases using LN-MFP and QTF.							
Target analyte	Detector type	Wavelength (μm)	IT (s)	Power (mW)	MDL (ppb)	RMDL ($\text{ppm}\cdot\text{W}$)	NNEA ($\text{cm}^{-1}\cdot\text{W}\cdot\text{Hz}^{-1/2}$)
NO ₂	LN-MFP	0.45	1000	1200	2	2.4×10^{-3}	/
	QTF [7]	0.45	120	47	21	9.87×10^{-4}	/
H ₂ O	LN-MFP	1.39	6300	19	25	4.75×10^{-4}	5.82×10^{-9}
	QTF [8]	1.39	1	/	2000	/	1.2×10^{-8}
C ₂ H ₂	LN-MFP	1.53	1000	3700	3.5	1.3×10^{-2}	2.7×10^{-8}
	QTF [9]	1.53	370	5	21	1.05×10^{-4}	/
CO ₂	LN-MFP	2	1900	10.6	350	3.71×10^{-3}	2.53×10^{-8}
	QTF [10]	2	365	5	2640	1.32×10^{-2}	2.5×10^{-8}
CH ₄	LN-MFP	3.3	1200	6.5	2.5	1.63×10^{-5}	8.36×10^{-9}
	QTF [11]	3.3	1	/	50	/	2.9×10^{-9}
NH ₃	LN-MFP	9.7	1100	82.3	0.35	2.88×10^{-5}	6.58×10^{-10}

NNEA, normalized noise equivalent absorption coefficient; LN-MFP: lithium niobate multi-functional platform; IT: Integration time; QTF: quartz tuning fork; MDL: minimum detection limit; RMDL: relative minimum detection limit

LN-MFP's competitive photoacoustic performance arises from several complementary physical and practical factors. First, single-crystal lithium niobate exhibits substantially higher effective piezoelectric coupling (for appropriate crystal orientations and tensor components) than α -quartz, enabling LiNbO₃ devices to generate stronger electrical signals—and thus higher signal-to-noise ratios—for a given mechanical strain [5]. Second, the device geometry and resonant frequency were carefully optimized to balance acoustic coupling, thermoelastic drive and readout bandwidths ensuring robust multimodal transduction even for slowly relaxing species such as CO₂. Finally, lithium niobate's mature and rapidly evolving photonics ecosystem, featuring broad optical transparency, strong electro-optic properties and wafer-level processing, facilitates the integration of on-chip optics, waveguides and micro-collimation components that are challenging to implement in conventional quartz tuning-fork systems [6]. The chip-scale monolithic LN-MFP also minimizes parasitic capacitance and interconnect losses, enabling compact low-noise electronics near the transducer, thereby lowering the noise floor and enhancing NNEA.

Table 3s Photodetector using LN-MFP and pyroelectric sensors.				
Detector type	Characterized wavelength range	NEP (W/√Hz)	Response time	Responsivity (V/W)
LN-MFP	450–9770 nm	5.65×10^{-8}	1 s	373
QTF [12]	1540 nm	/	1 s	47.4
LiNbO ₃ [13]	/	1.623×10^{-7}	1.9 s	/
LiNbO ₃ [14]	8–11 μm	/	29 ms	/
LiTaO ₃ [15]	X-ray	5.02×10^{-8}	/	1800
Sb ₂ Te ₃ –Bi ₂ Te ₃ [16]	600–700 nm	8.0×10^{-9}	341 μs	38

LN-MFP: lithium niobate multi-functional platform; QTF: quartz tuning fork.

A concise comparison of the LN-MFP with representative pyroelectric and tuning-fork detectors is provided in Table 3s. In the current prototype the LN-MFP demonstrates high responsivity ($\sim 373 \text{ V W}^{-1}$) and a noise-equivalent power on the order of $5.6 \times 10^{-8} \text{ W Hz}^{-1/2}$; measured with a 1 s lock-in time constant (corresponding to $\sim 0.25 \text{ Hz}$ detection bandwidth). The device operates across a broad wavelength range from 450 nm to 9.77 μm and integrates PAS, LITES and direct photodetection functionalities on a single chip. The reported performance reflects a deliberate, system-level optimization, balancing geometry and readout to meet the combined requirements of all three sensing modalities. Although the LN-MFP is not necessarily optimized for any single modality, it uniquely combines high per-power responsivity, multimodal functionality and compatibility with wafer-scale photonics. Several straightforward engineering routes remain to further improve the absolute MDL, NEP and bandwidth, including the use of low-noise transimpedance amplifiers, micro-collimation or waveguide coupling to enhance optical power delivery, and tailored absorptive coatings at the fine base. These improvements will be explored in future work to further enhance performance while preserving the platform’s integration advantages.

Table 4s LITES detection of trace gases using LN-MFP and QTF.						
Target analyte	Detector type	Wavelength (μm)	IT (s)	AL (cm)	MDL (ppm)	RMDL ($\text{ppm}\cdot\text{m}$)
NO ₂	LN-MFP	0.45	450	2.5	1	2.5×10^{-2}
H ₂ O	LN-MFP	1.39	1600	2.5	2.5	6.25×10^{-2}
	QTF [17]	1.39	289	50	1.4	7×10^{-1}
C ₂ H ₂	LN-MFP	1.53	1500	2.5	0.45	1.13×10^{-2}
	QTF [18]	1.53	70	20	0.36	7.2×10^{-2}
CO ₂	LN-MFP	2	1200	2.5	35	8.75×10^{-1}
	QTF [19]	2	100	100	20	2×10^1
CH ₄	LN-MFP	3.3	1900	2.5	0.9	2.25×10^{-2}
	QTF [20]	1.65	206	420	0.0035	1.47×10^{-2}
NH ₃	LN-MFP	9.7	1200	2.5	1.2	3×10^{-2}

AL: Absorption Length; QTF: quartz tuning fork; LN-MFP: lithium niobate multi-functional platform; MDL: minimum detection limit; RMDL: relative minimum detection limit; IT: Integration time;

For LITES, the key parameters include the laser wavelength, absorption length (AL), and integration time, as summarized in Table 4s. Since the MDL is directly proportional to the AL, the comparison includes not only the MDL but also the relative MDL for a fair and comprehensive evaluation. Table 4s shows that the LN-MFP attains superior RMDL for most channels (H₂O, C₂H₂, CO₂, NO₂, NH₃), with CH₄ as the only minor exception. This advantage stems from three factors: larger effective piezoelectric coupling of Y+128° LiNbO₃, an electrode layout that maximizes charge collection at regions of peak modal strain, and a resonance (~10.5 kHz) deliberately tuned to enhance thermoelastic/acoustic energy build-up for slowly relaxing species.

Reference

5. Yue, W. et al. Crystal orientation dependence of piezoelectric properties in LiNbO₃ and LiTaO₃. *Opt. Mater.* **23**, 403-408 (2003).
6. Boes, A. et al. Lithium niobate photonics: Unlocking the electromagnetic spectrum. *Science* **379**, eabj4396 (2023).
7. Breitegger, P. et al. Towards low-cost QEPAS sensors for nitrogen dioxide detection. *Photoacoustics* **18**, 100169 (2020).

8. Rousseau R. et al. Monolithic double resonator for quartz enhanced photoacoustic spectroscopy. *Appl. Sci.* **11**: 2094 (2021).
9. Wang, R., Qiao, S., He, Y., & Ma, Y. Highly sensitive laser spectroscopy sensing based on a novel four-prong quartz tuning fork. *Opto-Electron. Adv.* **8**, 240275-1 (2025).
10. Zhang, Y. et al. Continuous real-time monitoring of carbon dioxide emitted from human skin by quartz-enhanced photoacoustic spectroscopy. *Photoacoustics* **30**, 100488 (2023).
11. Wu H. et al. Atmospheric CH₄ measurement near a landfill using an ICL-based QEPAS sensor with VT relaxation self-calibration. *Sens. Actuator B-Chem.* **297** 126753 (2019).
12. Zhou, S. et al. Realization of a infrared detector free of bandwidth limit based on quartz crystal tuning fork. *Opt. Laser Technol.* **113**, 261-265 (2019).
13. Aleks, M. et al. An overview of microelectronic infrared pyroelectric detector. *Eng. Sci.* **16**, 82-89 (2021).
14. Suen, J. Y. et al. Multifunctional metamaterial pyroelectric infrared detectors. *Optica* **4**, 276-279 (2017).
15. Kane, S. R. et al. Characterizing pyroelectric detectors for quantitative synchrotron radiation measurements. *Sensor. Actuat. A-Phys.* **387**, 116406 (2025).
16. Mauser, K. W. et al. Resonant thermoelectric nanophotonics. *Nat. Nanotechnol.* **12**, 770-775 (2017).
17. Xu, L. et al. Multigas sensing technique based on quartz crystal tuning fork-enhanced laser spectroscopy. *Anal. Chem.* **92** 14153-14163 (2020).
18. He, Y. et al. Hydrogen-enhanced light-induced thermoelastic spectroscopy sensing. *Photonics Res.* **13** 194-200 (2024).
19. Bojęś, P. et al. Dual-band light-induced thermoelastic spectroscopy utilizing an antiresonant hollow-core fiber-based gas absorption cell. *Appl. Phys. B* **129**, 177 (2023).
20. Shang, Z. et al. Robust and compact light-induced thermoelastic sensor for atmospheric methane detection based on a vacuum-sealed subminiature tuning fork. *Photoacoustics* **42**, 100691 (2025).”

Reviewer #2:

The manuscript describes a lithium niobate multifunctional integrated platform (LN-MFP) for spectroscopic gas sensing. Specifically, the authors present a systematic study of a lithium niobate tuning fork for photoacoustic, thermoelastic/photodetection-based sensing. It is of interest to see the direct bonding of the QCL chip to the LN-MFP, showing potential for integrated spectroscopic applications. This work will attract researchers from different areas such as spectroscopy, sensing, integrated photonics, etc. The following issues should be addressed before this work can be considered for publication.

1. One key application shown in this study is photoacoustic spectroscopy. The authors show a good record of adopting tuning forks (especially quartz tuning fork, QTF) in gas sensing with demonstrated high sensitivity across a broad spectral range. However, in the manuscript, the authors seem to only report the PAS gas sensing results without providing sufficient discussion. A detailed comparison with state-of-the-art tuning fork-based PAS gas sensing systems is needed to highlight the key advantages and advancements offered by the designed LN tuning fork.

Thank you for the reviewer's valuable suggestion. A detailed comparison table has been added to the Supplementary.

In the main article (page 9 line 14), we have included the sentence:

“The photoacoustic detection performance of the LN-MFP and the QTF-based PAS systems is summarized in Supplementary Table 2s.”

In the Supplementary (page 14 line 2), we have included the following sentences:

“LN-MFP's competitive photoacoustic performance arises from several complementary physical and practical factors. First, single-crystal lithium niobate exhibits substantially higher effective piezoelectric coupling (for appropriate crystal orientations and tensor components) than α -quartz, enabling LiNbO₃ devices to generate stronger electrical signals—and thus higher signal-to-noise ratios—for a given mechanical strain [5]. Second, the device geometry and resonant frequency were carefully optimized to balance acoustic coupling, thermoelastic drive and readout bandwidths ensuring robust multimodal transduction even for slowly relaxing

species such as CO₂. Finally, lithium niobate’s mature and rapidly evolving photonics ecosystem, featuring broad optical transparency, strong electro-optic properties and wafer-level processing, facilitates the integration of on-chip optics, waveguides and micro-collimation components that are challenging to implement in conventional quartz tuning-fork systems [6]. The chip-scale monolithic LN-MFP also minimizes parasitic capacitance and interconnect losses, enabling compact low-noise electronics near the transducer, thereby lowering the noise floor and enhancing NNEA.”

Table 2s | PAS detection of trace gases using LN-MFP and QTF.

Target analyte	Detector type	Wavelength (μm)	IT (s)	Power (mW)	MDL (ppb)	RMDL (ppm·W)	NNEA (cm ⁻¹ ·W·Hz ^{-1/2})
NO ₂	LN-MFP	0.45	1000	1200	2	2.4×10 ⁻³	/
	QTF [7]	0.45	120	47	21	9.87×10 ⁻⁴	/
H ₂ O	LN-MFP	1.39	6300	19	25	4.75×10 ⁻⁴	5.82×10 ⁻⁹
	QTF [8]	1.39	1	/	2000	/	1.2×10 ⁻⁸
C ₂ H ₂	LN-MFP	1.53	1000	3700	3.5	1.3×10 ⁻²	2.7×10 ⁻⁸
	QTF [9]	1.53	370	5	21	1.05×10 ⁻⁴	/
CO ₂	LN-MFP	2	1900	10.6	350	3.71×10 ⁻³	2.53×10 ⁻⁸
	QTF [10]	2	365	5	2640	1.32×10 ⁻²	2.5×10 ⁻⁸
CH ₄	LN-MFP	3.3	1200	6.5	2.5	1.63×10 ⁻⁵	8.36×10 ⁻⁹
	QTF [11]	3.3	1	/	50	/	2.9×10 ⁻⁹
NH ₃	LN-MFP	9.7	1100	82.3	0.35	2.88×10 ⁻⁵	6.58×10 ⁻¹⁰

NNEA, normalized noise equivalent absorption coefficient; LN-MFP: lithium niobate multi-functional platform; IT: Integration time; QTF: quartz tuning fork; MDL: minimum detection limit; RMDL: relative minimum detection limit

Reference

- Yue, W. et al. Crystal orientation dependence of piezoelectric properties in LiNbO₃ and LiTaO₃. *Opt. Mater.* **23**, 403-408 (2003).
- Boes, A. et al. Lithium niobate photonics: Unlocking the electromagnetic spectrum. *Science* **379**, eabj4396 (2023).
- Breitegger, P. et al. Towards low-cost QEPAS sensors for nitrogen dioxide detection. *Photoacoustics* **18**, 100169 (2020).
- Rousseau R. et al. Monolithic double resonator for quartz enhanced photoacoustic spectroscopy. *Appl. Sci.* **11**: 2094 (2021).

9. Wang, R., Qiao, S., He, Y., & Ma, Y. Highly sensitive laser spectroscopy sensing based on a novel four-prong quartz tuning fork. *Opto-Electron. Adv.* **8**, 240275-1 (2025).
10. Zhang, Y. et al. Continuous real-time monitoring of carbon dioxide emitted from human skin by quartz-enhanced photoacoustic spectroscopy. *Photoacoustics* **30**, 100488 (2023).
11. Wu H. et al. Atmospheric CH₄ measurement near a landfill using an ICL-based QEPAS sensor with VT relaxation self-calibration. *Sens. Actuator B-Chem.* **297** 126753 (2019).”

2. In addition, the following information should be added to strengthen the technical content:

(1) Experimental setup for each light source is suggested to be documented in the supplement.

(2) Better to include N₂ background signals in Fig. 2 for direct comparison.

Thanks for the comment.

(1) In accordance with the reviewer's recommendation, the experimental setup for each light source is documented.

In the Supplementary page 18, we add a Table 5s:

Table 5s. The wavelength and model of seven light sources.						
Laser source	Model	Wavelength (nm)	Power (mW)	Modulation method	IC (mA)	Temperature (°C)
VIS LED	NICHIA 450	450	1200	AM	600	/
NIR LD	NTT Electronics Corporation	1390	19	FM	84	28
NIR LD	NTT Electronics Corporation	1536	3700*	FM	90	25
NIR LD	NTT Electronics Corporation	2004	10.6	FM	120	18
MWIR ICL	NANOPLUS	3370	6.5	FM	86	17
QCL Chip	Chinese Academy of Sciences	4590	14	FM	270	45
LWIR QCL	HEATHYPHOTON	9770	82.3	FM	420	20

IC: injection current; AM: amplitude modulation; FM: frequency modulation; *Amplified by EDFA

In addition, in the Supplementary Page 16 line 8, we add:

“Figure 7s shows a photo of the experimental setup used for the LN-MFP measurements. A

function generator (Tektronix AFG3102) supplied the modulation waveform to the light sources. The sources, as listed in the Table 5s, were driven and temperature-stabilized by a current/TEC controller (Thorlabs ITC4005QCL). The emitted beam was routed through the detection module. A camera (Dataray WinCamD-IR-BB) was employed for beam profiling and alignment. The electrical output from the detection module was demodulated by a lock-in amplifier (SRS SR830). A personal computer was used for instrument control and data acquisition.’

Fig. 7s | Photo of the experimental setup. PC: personal computer, DM: detection module.

(2) Thanks for the valuable suggestion. We have already included N_2 background signals in Fig. 2, as shown in the revised main article, page 9.

Fig. 2 | LN-MFP for photoacoustic detection. **a** Spectral coverage with excitation sources from visible to long-wave infrared and the targeted analytes. **b-g** present representative measurements for each target gas: the main plot in each panel shows a typical $2f$ signal acquired with the indicated light source and sample concentration; the left inset shows the concentration-calibration (linear fit); the top-right inset shows a 1σ noise level obtained with pure N_2 ; and the bottom-right inset shows the Allan-deviation curves used to extract the 1σ minimum detection limit (MDL) and optimal averaging times. Extracted MDLs (1σ) and averaging times are: **b** MDL ~ 2 ppb at 1,000 s; **c** MDL ~ 25 ppb at 6,300 s; **d** MDL ~ 3.5 ppb at 1,000 s; **e** MDL ~ 350 ppb at $\sim 1,900$ s; **f** MDL ~ 2.5 ppb at 1,200 s; **g** MDL ~ 350 ppt at 1,100 s. VIS LED: visible light emitting diode; NIR LD: near-infrared laser diode; MWIR ICL: mid-wave interband cascade laser; LWIR QCL: long-wave quantum-cascade laser.

3. Regarding the demonstration of thermoelastic detection (LITES), the reviewer notices tuning forks have achieved high sensitivity in the visible, near-IR, and mid-IR. The manuscript should compare with state-of-the-art LITES results and clarify how the LN tuning fork advances beyond these results.

Thanks for the comment. The comparisons with state-of-the-art LITES results have been added as Table 4s to the Supplementary page 16, as follows:

Table 4s LITES detection of trace gases using LN-MFP and QTF.						
Target analyte	Detector type	Wavelength (μm)	IT (s)	AL (cm)	MDL (ppm)	RMDL (ppm·m)
NO ₂	LN-MFP	0.45	450	2.5	1	2.5×10 ⁻²
H ₂ O	LN-MFP	1.39	1600	2.5	2.5	6.25×10 ⁻²
	QTF [17]	1.39	289	50	1.4	7×10 ⁻¹
C ₂ H ₂	LN-MFP	1.53	1500	2.5	0.45	1.13×10 ⁻²
	QTF [18]	1.53	70	20	0.36	7.2×10 ⁻²
CO ₂	LN-MFP	2	1200	2.5	35	8.75×10 ⁻¹
	QTF [19]	2	100	100	20	2×10 ¹
CH ₄	LN-MFP	3.3	1900	2.5	0.9	2.25×10 ⁻²
	QTF [20]	1.65	206	420	0.0035	1.47×10 ⁻²
NH ₃	LN-MFP	9.7	1200	2.5	1.2	3×10 ⁻²

AL: Absorption Length; QTF: quartz tuning fork; LN-MFP: lithium niobate multi-functional platform; MDL: minimum detection limit; RMDL: relative minimum detection limit; IT: Integration time;

Examining the RMDL entries in Table 4s shows that the LN-MFP delivers better RMDL for most tested channels (H₂O, C₂H₂, CO₂, NO₂, NH₃) compared with the cited QTF results. The only channel in the table where a cited QTF entry currently shows a slightly better RMDL is CH₄.

The superior RMDL of the LN-MFP arises from several complementary factors:

(1) Higher piezoelectric transduction per unit strain

For the relevant device orientations, single-crystal LiNbO₃ provides larger effective piezoelectric coupling than α-quartz. Consequently, for the same local thermoelastic or acoustic strain the LN prongs generate more collected charge at the electrodes, improving the electrical SNR and lowering RMDL.

(2) Optimized electrode geometry and charge collection

The LN-MFP electrode layout was properly designed to collect charge from regions of maximum modal strain (near the tine base). This increases the fraction of mechanically generated charge that is measured versus many standard QTF electrode patterns, improving per-power readout efficiency.

(3) Resonant tuning matches thermoelastic / molecular dynamics

The chosen resonance (~10.5 kHz) and geometry were selected to maximize thermoelastic/ acoustic energy accumulation for slowly relaxing species (e.g., CO₂). This resonance matching increases the thermoelastic drive per unit delivered optical power for such species and thus yields better RMDL in those channels.

In the Supplementary page 15 line 7, we added the following clarification:

“Table 4s shows that the LN-MFP attains superior RMDL for most gas species (H₂O, C₂H₂, CO₂, NO₂, NH₃), with CH₄ as the only minor exception. This advantage stems from three factors: larger effective piezoelectric coupling of Y+128° LiNbO₃, an electrode layout that maximizes charge collection at regions of peak modal strain, and a resonance (~10.5 kHz) properly tuned to enhance thermoelastic/acoustic energy build-up for slowly relaxing species.”

Reference

17. Xu, L. et al. Multigas sensing technique based on quartz crystal tuning fork-enhanced laser spectroscopy. *Anal. Chem.* **92** 14153-14163 (2020).
18. He, Y. et al. Hydrogen-enhanced light-induced thermoelastic spectroscopy sensing. *Photonics Res.* **13** 194-200 (2024).
19. Bojęś, P. et al. Dual-band light-induced thermoelastic spectroscopy utilizing an antiresonant hollow-core fiber-based gas absorption cell. *Appl. Phys. B* **129**, 177 (2023).
20. Shang, Z. et al. Robust and compact light-induced thermoelastic sensor for atmospheric methane detection based on a vacuum-sealed subminiature tuning fork. *Photoacoustics* **42**, 100691 (2025).

4. Following the previous question, some specific concerns:

(1) The claim that "resonant amplification significantly boosts sensitivity" is confusing, as the LITES performance is notably worse than PAS. It is noticed that the author used a 2.5 cm

optical length, but the results are still unsatisfied to support your claims. Further discussion is needed to clarify this discrepancy.

(2) Clarify how dual PAS/LITES functionality enables comprehensive analysis. It is not straightforward for readers to understand how combining them together can leverage the sensing power of LN-MFP.

(3) Clarify the relationship between spectral selectivity and dual functionality, as selectivity is primarily determined by ro-vibrational fingerprints.

Thanks for the comment.

(1) To avoid the confusion, we deleted the sentence:

“Notably, this resonant amplification effect significantly boosts the sensitivity of the system.”

(2) The integration of these two complementary detection modes enables comprehensive and versatile gas sensing under various environmental conditions.

Specifically, in the PAS mode, gas molecules are excited by a modulated laser beam that passes through the narrow tine gap of the LN-MFP. The resulting acoustic pressure is mechanically amplified by the resonant vibration of the fork structure. In photoacoustic detection, the analyte gas surrounds the LN-MFP, enabling high-sensitivity detection within a compact volume. However, when the target gas is corrosive or contains impurities, the LN-MFP may suffer from surface degradation or contamination, leading to reduced performance.

In contrast, when operated in the LITES mode, the LN-MFP functions in a non-contact detection configuration. In this case, the laser passes through the target gas, which can be enclosed in a sealed chamber if corrosive or contains for example particulates, and then irradiates the surface of the LN-MFP. The light induced thermoelastic deformation generates a measurable signal. Unlike photoacoustic processes relying on acoustic wave propagation through gas, this mechanism eliminates direct contact between gas and transducer. Consequently, LITES enables reliable detection even in extreme or corrosive environments where physical interaction between the gas and transducer could otherwise degrade performance.

By integrating both PAS and LITES modes on the same LN-MFP platform, the device can

adaptively switch between a compact PAS configuration and a non-contact LITES configuration, depending on the target gas and environmental conditions. This dual-mode functionality enables the system to accommodate a wide range of applications, from trace gas analysis with only minute gas volumes available to reliable monitoring under harsh industrial or atmospheric conditions. There by achieving comprehensive and flexible sensing capability within a unified, miniaturized device architecture.

To make it clear to the readers, in the revised main article in page 17 line 7, we replaced the original paragraph:

“The integration of thermoelastic detection into the LN-MFP demonstrates its versatility as a multifunctional spectroscopic platform. The capability to perform both photoacoustic and thermoelastic measurements within a single device highlights its potential for comprehensive spectroscopic analysis. Such dual functionality proves especially valuable for applications requiring both high sensitivity and spectral selectivity across a broad spectral range, further positioning the LN-MFP as an advanced tool for next-generation spectroscopic sensing technologies.”

with the following:

“The integration of thermoelastic detection into the LN-MFP demonstrates its versatility as a multifunctional spectroscopic platform. The spectral selectivity of both modes arises from the intrinsic ro-vibrational fingerprints of target molecules. The dual functionality does not alter this selectivity but extends its applicability across varying gases and conditions. By incorporating both PAS and LITES detection modes, the device enables adaptive gas sensing across a wide range of environments and measurement scenarios. In PAS mode, direct gas–sensor interaction provides high-sensitivity detection within a compact volume, ideal for confined or small-scale applications. Conversely, LITES operates in a non-contact configuration, suitable for sealed, corrosive or otherwise harsh environments. This dual-mode architecture combines the sensitivity and compactness of contact-based detection with the robustness and flexibility of non-contact operation, enabling comprehensive spectroscopic analysis and positioning the LN-MFP as a versatile platform for next-generation sensing technologies.”

(3) The spectral selectivity of the LN-MFP system originates from the intrinsic ro-vibrational fingerprints of target molecules, which fundamentally govern the absorption characteristics in both PAS and LITES. Since both techniques rely on absorption-based signal generation, their selectivity is inherently molecular rather than device-dependent. The dual PAS/LITES functionality does not modify this intrinsic selectivity. Instead, it broadens the operational scope of the system. In the PAS mode, direct gas–sensor interaction allows high-sensitivity detection of strong absorption lines within confined volumes, while the LITES mode enables non-contact measurements of weaker transitions or corrosive gases over larger or sealed spaces. Moreover, the broadband optical transparency and mechanical stability of the LN-MFP support excitation from the near- to mid-infrared, granting access to multiple absorption bands. Consequently, the dual-mode architecture preserves the molecular origin of spectral selectivity while extending its practical applicability across diverse gases, spectral regions, and environmental conditions.

To make it clear to readers, in main article page 17 line 7, we added the following sentences:

“The integration of thermoelastic detection into the LN-MFP demonstrates its versatility as a multifunctional spectroscopic platform. The spectral selectivity of both modes arises from the intrinsic ro-vibrational fingerprints of target molecules. The dual functionality does not alter this selectivity but extends its applicability across varying gases and conditions.”

5. It seems there is an overlap between the photodetection functionality with the thermoelastic sensing mechanism. Is it more reasonable to move this section before LITES as it relates to the fundamental mechanism? In addition, the modulated photodetection using tuning forks has been previously demonstrated, so it is more interesting to make a comparison with state-of-art photodetection using tuning forks, highlighting key pros and cons. Also, it is better to make a fair comparison of noise-equivalent power (NEP) with commercial photodetectors, which is more important in photodetection.

Thanks for the comment.

*(1) We have moved section “**Photodetection with the LN-MFP**” before the section*

“Thermoelastic detection with the LN-MFP”.

(2) We have added Table 3S to supplementary page 15, comparing the LN-MFP with representative pyroelectric and tuning-fork detectors.

Table 3s Photodetector using LN-MFP and pyroelectric sensors.				
Detector type	Characterized wavelength range	NEP (W/√Hz)	Response time	Responsivity (V/W)
LN-MFP	450–9770 nm	5.65×10^{-8}	1 s	373
QTF [12]	1540 nm	/	1 s	47.4
LiNbO ₃ [13]	/	1.623×10^{-7}	1.9 s	/
LiNbO ₃ [14]	8–11 μm	/	29 ms	/
LiTaO ₃ [15]	X-rap	5.02×10^{-8}	/	1800
Sb ₂ Te ₃ –Bi ₂ Te ₃ [16]	600–700 nm	8.0×10^{-9}	341 μs	38

LN-MFP: lithium niobate multi-functional platform; QTF: quartz tuning fork.

Reference

12. Zhou, S. et al. Realization of a infrared detector free of bandwidth limit based on quartz crystal tuning fork. *Opt. Laser Technol.* **113**, 261-265 (2019).

13. Aleks, M. et al. An overview of microelectronic infrared pyroelectric detector. *Eng. Sci.* **16**, 82-89 (2021).

14. Suen, J. Y. et al. Multifunctional metamaterial pyroelectric infrared detectors. *Optica* **4**, 276-279 (2017).

15. Kane, S. R. et al. Characterizing pyroelectric detectors for quantitative synchrotron radiation measurements. *Sensor. Actuat. A-Phys.* **387**, 116406 (2025).

16. Mauser, K. W. et al. Resonant thermoelectric nanophotonics. *Nat. Nanotechnol.* **12**, 770-775 (2017).

In the present prototype the LN-MFP exhibits high responsivity (~373 V/W) and noise-equivalent power on the order of 5.6×10^{-8} W/√Hz; these values were measured with a 1 s lock-in time constant (~0.25 Hz detection bandwidth). The device supports a characterized

wavelength range from 450 nm to 9.77 μm and integrates PAS, LITES and direct photodetection modalities on a single chip. It is important to note that the reported performance represents a deliberate, system-level compromise: the geometry and readout were optimized to satisfy the requirements of all three sensing modalities. While the LN-MFP is not necessarily optimized for any single sensing modality, it achieves a balanced performance across all functions. Its combination of high per-power responsivity, multimodal capability, and compatibility with wafer-scale photonics distinguishes it from existing devices. Moreover, several straightforward engineering improvements remain to enhance the absolute MDL, NEP, and bandwidth—such as implementing low-noise transimpedance amplifiers, integrating micro-collimators or waveguide coupling to improve optical power delivery, and applying tailored absorptive coatings at the tine base. These developments will be pursued in future work to further advance each modality while preserving the platform's high level of integration.

(3) A fair comparison of the noise-equivalent power (NEP) between the LN-MFP and the commercial photodetector has been included.

In the main article page 13, we replaced the original Table 3 with the following:

Table 2 LN-MFP for photodetection from VIS to LWIR.					
Detector type	Light sources	Wavelength (μm)	MDL (nW)	NEP (W/√Hz)	Photoresponsivity (V/W)
LN-MFP	VIS LED	0.45	260	5.88×10^{-7}	67.7
		1.39	445	1×10^{-6}	35.5
	NIR LD	1.53	695	1.57×10^{-6}	22.5
		2	693	1.56×10^{-6}	15.3
	MIR ICL	3.37	1000	2.26×10^{-6}	9.8
	LWIR QCL	9.77	25	5.65×10^{-8}	373
HPPD-B-D-10.6-10	LWIR QCL	9.77	8000	1.8×10^{-5}	258

6. LN is an important photonic platform used in many areas as described in the first paragraph of Introduction. However, the authors missed mentioning the spectroscopic sensing applications of using LN. May be not too many in this topic, but at least there are a couple of pioneering work recently reported on fabricating LN rib waveguide and using it for on-chip photothermal gas sensing. As spectroscopic sensing the key topic of this paper, the authors should make a more complete literature to strengthen this part.

Additional concerns:

- (1) Why is the tuning fork placed horizontally rather than vertically?
- (2) It is more interesting to include a photo/microscope image of the on-chip LN-MFP sensor in Figure 5.
- (3) The results show the signal of 8000 ppm CO. Please discuss limitations and potential solutions.

Thanks for the comment. We have added the most recent publications on spectroscopic sensing in the main article page 3 line 4.

“Recent breakthroughs in LN photonics have propelled the development in high-performance

components, including electro-optic modulators [3-5], frequency shifters [6], frequency combs [7-9], pulse generators [10], tunable lasers [11], spectroscopic sensing [12-14] and photonic integrated circuits [15].”

Reference

[12] Shim, J. et al. Room-temperature waveguide-integrated photodetector using bolometric effect for mid-infrared spectroscopy applications. *Light Sci. Appl.* **14**, 125 (2025).

[13] Hwang, A. Y. et al. Mid-infrared spectroscopy with a broadly tunable thin-film lithium niobate optical parametric oscillator. *Optica* **10**, 1535-1542 (2023).

[14] Yan, Y. et al. On-chip photothermal gas sensor based on a lithium niobate rib waveguide. *Sensor Actuat. B-chem.* **405**, 135392 (2024).

Additional concerns:

(1) As shown in Fig. 1, the LN-MFP has a total length of approximately 1.5 cm. During the on-chip fabrication process, the device was positioned horizontally on the silicon substrate, primarily due to chip-size limitations and packaging considerations. The LN-MFP operates through the resonant vibration of its tines, a mechanism independent of its overall orientation. To prevent mechanical damping and ensure free tine motion, a micromachined recess was created beneath the fork region using photolithography, allowing the tines to remain suspended without contacting the substrate. Moreover, the horizontal configuration enables in-plane optical detection, which increases the effective light–gas interaction length and improves detection sensitivity. Therefore, horizontal arrangement represents a practical and robust design choice that balances mechanical performance, optical efficiency, and integration feasibility.

(2) The photograph of the on-chip LN-MFP sensor was added to main article Figure 5.

Fig. 5 | On chip LN-MFP sensor. **a** Photograph of the packaged on-chip sensor. **b** Optical-microscope image of the coupling region showing the QCL emission facet aligned with the LN-MFP tine gap. **c** Schematic diagram of the free-space optical coupling. **d** $2f$ photoacoustic spectrum of 8,000 ppm CO in N₂ measured using the integrated LN-MFP–QCL assembly.

(3) As shown in the revised Fig. 5c, the laser emitted from the QCL diverges by approximately 70° across a ~100 μm gap without any collimating optics, causing part of the beam to illuminate the LN-MFP surface. This unintended illumination introduces additional photothermal noise, thereby degrading the signal-to-noise ratio. In future work, we plan to incorporate micro collimating optics or integrated LN waveguides to confine the laser propagation path and prevent stray light from reaching the tines. Such improvements are expected to significantly reduce optical noise, enhance coupling efficiency, and ultimately improve the detection limit and dynamic range of the on-chip LN-MFP sensor.

In the revised main article page 19 line 14, we replaced the sentences:

“Specifically, our goal is to merge a mid-infrared quantum cascade laser (QCL) chip with the LN-MFP using advanced chip fabrication techniques, thereby consolidating both the light

source and the detector onto a single platform.”

With the following:

“Specifically, our goal is to integrate a mid-infrared quantum cascade laser (QCL) chip and optical waveguide with the LN-MFP through advanced microfabrication techniques, thereby unifying the light source, waveguide, and detector on a single monolithic platform.”

7. Regarding the design and fabrication of the LN-MFP, a discussion and comparison with existing LN, quartz, silicon, and polymer tuning forks should be added. The key parameters (resonant frequency, quality factor) should be moved to the main text, with detailed simulation and characterization data properly referenced in supplementary materials. The key parameters like resonant frequency, quality factor, and dimensions will be useful for readers to understand the advancements of the proposed device.

Thanks for the constructive suggestion.

In accordance with the reviewer’s recommendation, we have substantially revised both the main text and the Supplementary to provide a detailed discussion of the LN-MFP’s design and fabrication.

(1) Discussion and comparison with existing tuning forks

We added the following discussion and comparison to the Supplementary in page 2:

“1. Discussion and Comparison Regarding the Design and Fabrication of Tuning Forks

The design and fabrication of tuning forks are strongly governed by the intrinsic material properties and the target application. The fabrication methods of representative tuning forks are summarized in Table 1s. Quartz tuning forks, typically produced by photolithography and wet chemical etching from Z-cut quartz wafers, exhibit excellent frequency stability, high mechanical Q-factor, and low temperature drift, making them ideal for precision sensing and timing applications. Silicon tuning forks, fabricated using photolithography and deep reactive ion etching (DRIE) within standard MEMS processes, offer high fabrication reproducibility and seamless compatibility with on-chip integration, however their lack of intrinsic piezoelectricity limits direct signal transduction. In contrast, LiNbO₃ tuning forks, fabricated by UV photolithography and optical-grade dicing, leverage lithium niobate’s strong

piezoelectric and electro-optic properties to enable efficient transduction and potential coupling with optical or acoustic fields. Nonetheless, their brittle crystalline nature and processing complexity impose strict requirements on machining precision. Polymer-based (PVDF) tuning fork, formed through film casting, mechanical shaping, and electrode deposition, provide a flexible, lightweight, and low-cost alternative. Despite their lower Q-factor and reduced frequency stability compared with crystalline counterparts, PVDF designs are advantageous for flexible, wearable, or chemically harsh sensing environments. Overall, the trade-off between precision, cost, mechanical robustness, and integration feasibility dictates the optimal material and fabrication strategy for a given tuning-fork application.”

Table 1s Design and fabrication method of existing tuning forks.			
Materials	Piezoelectric	Fabrication method	Ref
LiNBO3	yes	UV photolithography and optical grade dicing	This work
quartz	yes	photolithography and etching	[1]
silicon	no	photolithography and etching	[2]
polymer	yes	film casting and mechanical processing	[3]
aluminum	no	mechanical processing	[4]

References

1. Lin, H. et al. Application of standard and custom quartz tuning forks for quartz-enhanced photoacoustic spectroscopy gas sensing. *Appl. Spectrosc. Rev.* **58**, 562-584 (2023).
2. Lavrik, N. V. & Datskos., P. G. Optically read Coriolis vibratory gyroscope based on a silicon tuning fork. *Microsyst. Nanoeng.* **5**, 47 (2019).
3. Liu, K. et al. A novel photoacoustic spectroscopy gas sensor using a low cost polyvinylidene fluoride film. *Sens. Actuator B-Chem.* **277**, 571-575 (2018).
4. Pan, Y. et al. Miniaturized and highly-sensitive fiber-optic photoacoustic gas sensor based on an integrated tuning fork by mechanical processing with dual-prong differential measurement. *Photoacoustics* **34**, 100573 (2023).

(2) key parameters with detailed simulation

The key parameters including resonant frequency and quality factor has been added to the main article in page 5 line 23:

“The LN-MFP has a frequency of 10,485 Hz and a Q factor of 1,621.”

The detailed simulation and characterization data has been added to the Supplementary section 2 page 3 line 1:

“The resonance frequency and quality factor (Q) of the LN-MFP are governed by both material properties and geometric configuration. In this work, the geometry was carefully optimized to balance the competing requirements of its multifunctional operation, including PAS, LITES, and photodetection. To guide this optimization, we present a developed systematic design procedure detailing how the geometric parameters of the LN-MFP were selected and refined.

(1) Optimization for Resonance

The geometry and resonance of the LN-MFP were optimized by varying two key geometric parameters: the tine length l and width w , while keeping all other parameters fixed (see Fig. 1s a for definitions). A parametric finite-element sweep of l and w revealed that the fundamental resonance f_0 decreases monotonically with increasing l and with decreasing w (Fig. 1s b). In the simulated design space f_0 ranged from ~ 34.5 kHz ($l=6$ mm, $w=2$ mm) to ~ 2.8 kHz ($l=18$ mm, $w=1$ mm).

For the LN-MFP, excessively high resonant frequency f_0 yields shorter vibration period, limiting the time available for the device to accumulate acoustic energy. Since the light-source modulation must match the LN-MFP resonance frequency, high f_0 values require faster modulation rates, which reduce the efficiency of photoacoustic generation for molecules with long vibrational-to-translational (V–T) relaxation times. For example, CO₂ exhibits V–T relaxation times on the order of 100 μ s (~ 10 kHz), and modulation beyond this frequency significantly diminishes photoacoustic generation efficiency. By contrast, very low f_0 increases susceptibility to low-frequency environmental disturbances such as mechanical vibration, air currents and $1/f$ (pink) noise.

Balancing these effects, an optimal frequency $f_0 \sim 10.5$ kHz was selected, corresponding to tine dimensions of $l=11.5$ mm, $w=1.7$ mm.

Fig. 1s | **a** Schematic of the LN-MFP showing the defined geometric parameters: tine length l , base width W , tine width w , inter-tine gap g , thickness t , and fillet radius r . **b** Finite-element simulation of the fundamental resonant frequency f_0 as a function of tine length l and width w . The results show that f_0 decreases markedly with increasing l and with decreasing w ; spanning a frequency range from ~ 34.5 kHz to ~ 2.8 kHz across the design space.

(2) Optimization for PAS

The LN-MFP geometry was optimized for photoacoustic spectroscopy (PAS) through coupled acoustic–structural finite-element simulations. In the acoustic excitation model, a cylindrical pressure source was positioned within the tine gap to reproduce the spatial distribution of the photoacoustic pressure wave and to induce symmetric tine oscillation (see Fig. 2s a). For each geometric configuration the total piezoelectric charge generated by the device was calculated.

The resulting contour map of normalized piezoelectric charge exhibits a clear and monotonic dependence on tine dimensions (Fig. 2s b). The PAS signal strength increases with greater tine length l and with smaller tine width w . Physically, longer and more slender tines exhibit larger tip displacements and stronger strain localization near the tine base under acoustic excitation, hereby generating a higher piezoelectric charge output and enhancing signal amplitude.

Fig. 2s | **a** Simulated deformation and strain distribution of the LN-MFP under PAS; **b** Normalized piezoelectric charge as a function of tine length l and tine width w .

(3) Optimization for LITES and photodetection

Because both LITES and direct photodetection originate from light-induced heating, they were analyzed within a unified thermoelastic framework. In this model, a localized heat source was applied at the LN-MFP base to simulate the photothermal excitation induced by incident light (see Fig. 3s a). For each geometric configuration, the total piezoelectric charge collected by the electrode was computed. The resulting contour map of normalized piezoelectric charge in (Fig. 3s b) reveals a systematic increase with the tine width w and only a weak dependence on tine length l . This trend arises from several factors. First, wider tines provide a larger heated volume and absorption area for a given illumination spot, leading to greater overall thermal expansion and a larger bending moment. Second, increasing w enlarges the electromechanically active volume and electrode-covered area, enhancing conversion of localized strain into piezoelectric charge. Third, because the thermoelastic excitation is concentrated near the base, variations in tine length l exert minimal influence on the thermal-to-mechanical coupling in this region.

Fig. 3s | **a** Simulated deformation and strain distribution of the LN-MFP under LITES and photodetection excitation; **b** Contour map of normalized piezoelectric charge as a function of tine length l and tine width w .

(4) Other geometry parameters

-Tine gap g

To accommodate diverse optical sources, from visible LEDs to mid-IR QCLs, and to balance optical and acoustic requirements, the tine gap was fixed at $g=1.0$ mm. The gap size is governed by two competing effects. For LITES and photodetection, light is incident on the outer tine surfaces, whereas in PAS operation, the excitation beam must traverse the gap without contacting the electrodes. If the gap is too narrow, the laser spot overlaps the metalized tines, introducing significant photothermal background noise. Conversely, if the gap is too wide, the acoustic pressure field weakens, reducing acoustic-to-mechanical coupling efficiency. Therefore, a 1.0 mm gap represents a practical compromise between these competing optical and acoustic constraints.

- Device thickness t

Commercial lithium-niobate wafers are typically available in thicknesses of ~ 500 μm . Within the explored design-space, variation in device thickness produced only minor effects on the fundamental resonance compared to the dominant influence of tine length and width. Its impact on electromechanical and thermoelastic responses was similarly limited. Considering

factors such as wafer availability, fabrication precision, mechanical robustness and process repeatability, a thickness of $t=500 \mu\text{m}$ was selected for the fabricated LN-MFP devices.

(5) Conclusion

In summary, a comprehensive finite-element parameter sweep combined with multiphysics simulation was conducted to optimize the LN-MFP geometry across three key performance dimensions: resonant behavior, photoacoustic (PAS) response, and thermoelastic/photodetection response. The inherently opposing design requirements of PAS and LITES/photodetection necessitated a balanced configuration. The finalized structure, with $l=11.5 \text{ mm}$, $w=1.7 \text{ mm}$, $g=1.0\text{mm}$, and $t=500 \mu\text{m}$, exhibits a fundamental resonant frequency near 10.5 kHz. This optimized design achieves high PAS sensitivity alongside strong LITES/photodetection responsivity, ensuring multifunctional operation with excellent mechanical stability, fabrication compatibility and system integration compatibility.”

Reviewer #3:

The paper presents a new device made from lithium niobate (LiNbO_3) that can do three things: detect sound made by light (photoacoustic), sense heat changes (thermoelastic), and respond to light across a wide range from visible to long-wave infrared. The experiments show it could be useful for small integrated sensors. The authors used the special electric and optical properties of LiNbO_3 to build a small sensor that does multiple things, which is both creative and technically important. I think this paper could be published if some issues are fixed. Here are my concerns:

-The paper says they added an acoustic resonator around the LN-MFP to make the signal stronger. According to the Supporting Information, the signal became 79 times bigger. But how big is the resonator compared to the tuning fork? How are they placed relative to each other? Did they try to match the size of the resonator to the frequency of the tuning fork? Also, did they test or simulate how well the resonator and tuning fork work together?

We thank the reviewer for this insightful comment regarding the acoustic resonator design and its coupling with the LN-MFP. We have now provided detailed information and analysis in both the revised manuscript and the Supplementary Materials to clarify the geometry, placement, and resonance matching between the acoustic resonator and the LN-MFP.

First, to help readers understand the dimensions of the acoustic resonator, we have added the dimensions of the acoustic resonator description in the Supplementary page 10 line 18:

“After optimization, the acoustic resonator was designed with a length of 13 mm, an inner diameter of 2.6 mm, and an outer diameter of 3 mm.”

Second, regarding resonance matching, the acoustic resonator’s longitudinal dimension was designed to match the acoustic wavelength corresponding to the LN-MFP’s fundamental frequency (~10.5 kHz). This was intentionally done so that the standing acoustic wave inside the resonator couples efficiently to the mechanical resonance of the fork. As shown in the revised Fig. 5s(c), the introduction of the resonator shifted the system’s resonance from 10,485 Hz to 10,490 Hz, indicating acoustic–mechanical coupling between the resonator and the LN-MFP. The quality factor (Q) slightly decreased to ~1300 due to increased acoustic loading,

confirming that the two components are indeed coupled. We have added the description in the Supplementary page 10 line 20:

“According to Fig. 5s c, the resonator shifted the system’s resonance from 10,485 Hz to 10,488 Hz, indicating acoustic–mechanical coupling between the resonator and the LN-MFP. The quality factor decreased to ~1300 due to resonant coupling.”

In Supplementary page 10, we replaced original Fig. 2s with the following Fig. 5s:

Fig. 5s | The LN-MFP coupled with AmRs. a Simulated and measured PAS signals as a function of the laser beam position along the LN-MFP tines. **b** PAS signal amplitude as the function of an acoustic resonator’s length. **c** Resonance curves measured for a bare LN-MFP and a LN-MFP with an AmRs; **d** PAS $2f$ signal based on H_2O measurements when using a bare LN-MFP and a LN-MFP with an AmRs.

-In the thermoelastic detection part, the laser was pointed at the side of the tuning fork prong instead of the front. Why was that done? Does lighting from the side help create more thermal

stress or better vibration? Were any comparisons done between side and front illumination to see which works better?

We thank the reviewer for this insightful comment regarding the laser incident position. This enhancement arises because, when the laser irradiates the bottom side of the prong, the optical path inside the LN substrate is longer, leading to more effective light absorption and stronger localized heating near the base region where mechanical strain is maximized. This generates a larger thermoelastic deformation and, consequently, a stronger piezoelectric response. In contrast, front illumination produces a shallower absorption depth and a less efficient thermal-stress distribution.

Therefore, the bottom-side illumination geometry was selected for all LITES and photodetection measurements to achieve stronger thermal excitation, improved signal-to-noise ratio, and better mechanical-electrical conversion efficiency. To make it clear to the readers, in the Supplementary in page 11 line 14.:

“As shown in Fig. 6s a, we experimentally compared the photodetection responses when the laser irradiated either the bottom side (black curve) or the front surface (red curve) of the LN-MFP tine. The results clearly indicate that side illumination produces a stronger signal, with a peak amplitude more than twice that of front illumination.”

In Supplementary page 12, we replaced original Fig. 3s with the following Fig. 6s:

Fig. 6s | Optimization of the laser incident position. a LITES signal of the LN-MFP as a function of the laser beam position along the left tuning fork tine. **b** Normalized photodetection signal measurements when laser was incident to the side and to the front of the LN-MFP tine

surface.

-The paper shows simulations saying that the shape and position of the electrodes affect how the device vibrates. But there's no experiment to prove this. Have the authors made real devices with different electrode shapes and tested them, like checking their frequency response or how they vibrate?

To validate the simulation results regarding the effect of electrode geometry on the vibration performance of the LN-MFP, and in accordance with the reviewer's comments, we fabricated and tested three electrode configurations, as shown in Fig. 2r. The LN-MFP contains six electrodes on both sides. In Configuration I, the two side electrodes on one surface are connected to the center electrode on the other side, corresponding to the optimal simulated design. In Configuration II, only the two side electrodes are connected, while the center electrode remains open-circuited. In Configuration III, all three electrodes on one surface are connected. Frequency sweep experiments show that Configuration I achieves the highest resonant amplitude, while Configuration II shows a modest decrease due to the unconnected center electrode. Configuration III exhibits an amplitude drop of approximately 70% due to the cancellation caused by the opposite charges on the center and side electrodes. These results confirm the high agreement between experimental measurements and simulations, demonstrating that electrode configuration significantly influences the electromechanical coupling efficiency of the LN-MFP.

Fig. 1r *a* Three types of electrode distributions. Red and blue indicate positive and negative poles. Black indicates open circuit. *b* Normalized resonance amplitude of different electrode

distribution.

-The authors used a Y-cut 128° LiNbO_3 crystal but didn't explain why. Since LiNbO_3 has different properties depending on how it's cut (like Z-cut, X-cut, etc.), maybe this cut gives better piezoelectric effect or mechanical quality. Could the authors say what makes this cut better for their device? Did they compare it with other cuts in simulations or tests?

We thank the reviewers for their valuable questions. The choice of Y-shaped, 128° LiNbO_3 for the fabrication of LN-MFP was based on literature evidence and our finite element simulation results. LiNbO_3 is known for its strong anisotropy—different crystal facets exhibit distinct piezoelectric coefficients, dielectric constants, and elastic properties, all of which directly affect the electromechanical coupling efficiency and vibration characteristics of the tuning fork-like structure. Y-shaped, 128° LiNbO_3 has been widely used in magnetic field sensing, atomic force microscopy (AFM), and oil density and viscosity measurements.

Fig. 2r Charge distribution generated by the piezoelectric effect of different properties analyzed by FEM.

To verify the optimal orientation for LN-MFP, we performed a COMSOL Multiphysics simulation and compared the piezoelectric charge distribution of Z-cut, X-cut, Y-cut, and Y-cut 128° LiNbO_3 wafers, as shown in Figure 3r. The results clearly show that the Y-cut 128° configuration produces the highest surface charge density. Therefore, choosing Y-cut 128° to prepare LN-MFP is the best solution.

-The paper mentions using a lock-in amplifier to get the 2f signal, but doesn't say how it was

set up. Important settings like time constant, filter slope, reference phase, and bandwidth can affect the signal quality. Could the authors give more details about these settings?

We thank the reviewer for this valuable comment and the opportunity to clarify the lock-in detection configuration used in our measurements. All second-harmonic (2f) demodulation signals were acquired using a Stanford Research SR830 lock-in amplifier, which was synchronized with the sinusoidal reference signal generated by a Tektronix AFG3102C function generator. This generator provided both the modulation signal to the laser driver and the reference input to the lock-in amplifier, ensuring precise phase synchronization between optical modulation and signal demodulation.

For all measurements, the reference frequency was set to match the resonant frequency of the LN-MFP (≈ 10.48 kHz). The time constant was set to 300 ms with a 12 dB/oct filter slope, providing a good balance between noise suppression and temporal response. The reference phase was optimized manually for each measurement to maximize the in-phase (X) component of the 2f signal, typically around $0^\circ \pm 2^\circ$. The detection bandwidth was determined by the chosen time constant and modulation frequency, corresponding to approximately 0.5–1 Hz in effective bandwidth.

This configuration ensured a high signal-to-noise ratio and stable demodulation of weak photoacoustic and thermoelastic signals, while maintaining consistent measurement conditions across all PAS, LITES, and photodetection experiments. We explicitly report the lock-in amplifier settings time constant and filter slope, as well as the integration time.

In the main article page 25 Line 4, we added:

“The lock-in amplifier was configured with a 1 s time constant and a 12 dB/octave low-pass filter slope, corresponding to an effective detection bandwidth of 0.25 Hz.”

-In many sensing methods like QEPAS and LITES, quartz is usually used because it's stable and has good resonance. Why did the authors choose LiNbO₃ instead? Is it because it can do more than one thing, has better sensitivity, or can be used with light on a chip? A comparison with quartz would help explain this choice.

We appreciate the reviewer's insightful question. To provide a direct comparison between

quartz and LiNbO₃, two tuning-fork sensors with identical geometries and dimensions were fabricated using the same process described in the main text. Their resonance characteristics were then experimentally measured by point-scanning detection, and the results are presented in Fig. 4r.

As shown, the quartz-based sensor exhibits a resonant frequency approximately 700 Hz lower than that of the LiNbO₃-based sensor, mainly due to the intrinsic differences in density and Young's modulus between the two materials. Furthermore, because LiNbO₃ possesses a much higher piezoelectric coefficient (e.g., $d_{33} \sim 27$ pm/V compared to quartz's $d_{11} \sim 2.3$ pm/V), the resonance amplitude of the LiNbO₃ device is significantly higher, confirming stronger electromechanical coupling and improved signal transduction efficiency.

Beyond its superior piezoelectricity, LiNbO₃ also offers excellent electro-optic and nonlinear optical properties, enabling integration of photoacoustic, thermoelastic, and photodetection functionalities within a single platform. This multifunctional capability makes LiNbO₃ a far more versatile material than quartz, particularly for future on-chip photonic implementations where compact, multimodal sensing and optical integration are desired.

Fig. 3r Resonance curves measured of quartz and LiNbO₃ sensors of the same size.

-The paper combines photoacoustic, thermoelastic, and photodetection into one device, but it's not clear why all three are needed together. What advantages does this combination give? It would help to have a comparison with using them separately.

We thank the reviewer for this valuable comment and the opportunity to clarify the advantages

of integrating photoacoustic (PAS), light-induced thermoelastic (LITES), and photodetection functionalities within a single LN-MFP platform. The multifunctional design is not a simple combination of sensing mechanisms, but rather a strategic integration that enables adaptive, comprehensive, and highly stable sensing under a wide range of environmental conditions.

Specifically, in the PAS mode, the modulated laser beam passes through the narrow tine gap of the LN-MFP and excites gas molecules within the confined region. The resulting acoustic pressure waves are mechanically amplified by the resonant fork structure, providing high-sensitivity gas detection in compact volumes—especially when coupled with an acoustic micro-resonator. This makes PAS particularly suitable for trace-gas analysis in controlled or enclosed environments where acoustic coupling is strong.

In contrast, the LITES mode operates through non-contact thermal transduction. Here, the laser irradiates the bottom sidewall region of the LN-MFP rather than directly passing through the gas. The absorbed optical energy induces periodic thermoelastic deformation in the LN substrate, generating detectable signals even under extreme or corrosive conditions where direct gas–transducer contact could impair performance or reliability. Thus, LITES provides a robust alternative detection mechanism in harsh or contamination-prone environments.

By integrating both PAS and LITES modes on the same LN-MFP platform, the device can adaptively switch between a compact PAS configuration and a non-contact LITES configuration, depending on the target gas and environmental conditions. This dual-mode functionality enables the system to accommodate a wide range of applications, from trace gas analysis with only minute gas volumes available to reliable monitoring under harsh industrial or atmospheric conditions. There by achieving comprehensive and flexible sensing capability within a unified, miniaturized device architecture.

To make it clear to the readers, in the revised main article in page 17 line 11, we replaced the original paragraph:

“The integration of thermoelastic detection into the LN-MFP demonstrates its versatility as a multifunctional spectroscopic platform. The capability to perform both photoacoustic and thermoelastic measurements within a single device highlights its potential for comprehensive spectroscopic analysis. Such dual functionality proves especially valuable for applications

requiring both high sensitivity and spectral selectivity across a broad spectral range, further positioning the LN-MFP as an advanced tool for next-generation spectroscopic sensing technologies.”

with the following:

“The integration of thermoelastic detection into the LN-MFP demonstrates its versatility as a multifunctional spectroscopic platform. The spectral selectivity of both modes arises from the intrinsic ro-vibrational fingerprints of target molecules. The dual functionality does not alter this selectivity but extends its applicability across varying gases and conditions. By incorporating both PAS and LITES detection modes, the device enables adaptive gas sensing across a wide range of environments and measurement scenarios. In PAS mode, direct gas–sensor interaction provides high-sensitivity detection within a compact volume, ideal for confined or small-scale applications. Conversely, LITES operates in a non-contact configuration, suitable for sealed, corrosive or otherwise harsh environments. This dual-mode architecture combines the sensitivity and compactness of contact-based detection with the robustness and flexibility of non-contact operation, enabling comprehensive spectroscopic analysis and positioning the LN-MFP as a versatile platform for next-generation sensing technologies.”

The built-in photodetection functionality further strengthens the system by enabling real-time monitoring of laser power, wavelength stability, and signal referencing—functions typically requiring separate detectors. This capability ensures self-calibration and enhances the stability and accuracy of both PAS and LITES measurements.

By integrating PAS, LITES, and photodetection within the same LN-MFP device, the system can flexibly switch between high-sensitivity PAS and robust LITES detection, while maintaining real-time optical feedback and system diagnostics. This tri-modal configuration expands the operational versatility of the device—from high-precision laboratory gas spectroscopy to industrial, atmospheric, or corrosive environment monitoring—all within a unified and miniaturized platform.

-For the photodetection tests, the authors changed the light power to see how the device

responds. But since LiNbO₃ absorbs different wavelengths differently, how do they know the change in signal is only because of power and not wavelength? Were the measurements done at fixed wavelengths? Or were power and wavelength controlled separately?

We thank the reviewer for this insightful question. All photodetection experiments in this study were conducted under fixed-wavelength conditions to ensure that the observed signal variations originated solely from changes in optical power rather than wavelength-dependent absorption.

Specifically, for each of the six laser sources (covering the visible to mid-infrared range), the output power was modulated electrically by applying a square-wave signal with a 50% duty cycle and 100% modulation depth to the injection current of the laser diode. This modulation scheme enabled periodic variation of the emitted optical power at a constant wavelength, allowing accurate characterization of the LN-MFP's photodetection response to power changes.

Although LiNbO₃ exhibits wavelength-dependent optical absorption, in the current experiments each test was performed using a single, fixed laser wavelength, corresponding to the absorption features of specific target gases. The differences among wavelengths were investigated by employing different laser sources (450 nm–9.77 μm) rather than by tuning a single laser across a wide spectral range. Therefore, under each measurement condition, the detected signal variations were attributed exclusively to changes in optical power, not to wavelength drift.

-In the conclusion, the authors talk about making a small photonic chip from this platform. But how will they turn this lab-scale device into something small enough for a chip? There are challenges like tiny fabrication, connecting parts on a wafer, and packaging. Also, how will they add light sources directly onto the same chip?

According to the reviewer's suggestion, we have added a detailed description in the Supplementary page 8 line16 section 3:

"The chip fabrication process began with the design and preparation of a transimpedance preamplifier PCB board, which serves as the signal conditioning and amplification interface for

the LN-MFP device. A gold-coated silicon substrate was then patterned using photolithography to define the electrode layout. To enable free vibration of the LN-MFP tines, a micro-machined recess was etched beneath the LN-MFP region on the silicon substrate, ensuring that the fork structure remained suspended and mechanically isolated from the base.

Subsequently, the processed silicon substrate was bonded to the preamplifier PCB, establishing the mechanical foundation for electrical integration. The LN-MFP was then attached to the silicon substrate using conductive silver paste, forming a robust electrical contact between the LN-MFP's bottom electrode and the silicon-based electrodes. In parallel, the QCL chip was mounted onto the same silicon substrate using a specialized adhesive or solder material, enabling optical alignment with the LN-MFP region.

Finally, gold wire bonding was employed to complete the electrical interconnections. The surface electrodes of the LN-MFP and QCL chip were bonded to the silicon electrodes, which were in turn connected to the corresponding pads on the PCB. This integration process ensured precise electrical continuity, compact structural assembly, and efficient signal transmission, providing the foundation for a stable and miniaturized chip-scale LN-MFP sensing system."

-The paper introduces a new multi-purpose LiNbO_3 platform, but it doesn't compare it clearly with existing technologies like QEPAS or LITES. Could the authors give a table or explanation comparing key performance numbers like detection limit, speed, and noise equivalent power?

Many thanks for the comment. Three detailed tables about PAS, LITES, and pyroelectric sensors were added to the Supplementary page 13 Section 9:

“9. Comparison of the performance of the LN-MFP with existing PAS, LITES, and pyroelectric sensors

“In photoacoustic spectroscopy, the key parameters that govern detection performance are the laser wavelength, optical power, and integration time. These parameters together with the minimum detection level (MDL), relative MDL (accounting for the proportionality of the photoacoustic signal to laser power), and normalized noise equivalent absorption (NNEA) coefficient, are summarized in Table 2s. Notably, the LN-MFP sensor is chip-scale offering inherent advantages in compactness, robustness, and ease of system integration.

Table 2s PAS detection of trace gases using LN-MFP and QTF.							
Target analyte	Detector type	Wavelength (μm)	IT (s)	Power (mW)	MDL (ppb)	RMDL (ppm·W)	NNEA (cm⁻¹·W·Hz^{-1/2})
NO ₂	LN-MFP	0.45	1000	1200	2	2.4×10 ⁻³	/
	QTF [7]	0.45	120	47	21	9.87×10 ⁻⁴	/
H ₂ O	LN-MFP	1.39	6300	19	25	4.75×10 ⁻⁴	5.82×10 ⁻⁹
	QTF [8]	1.39	1	/	2000	/	1.2×10 ⁻⁸
C ₂ H ₂	LN-MFP	1.53	1000	3700	3.5	1.3×10 ⁻²	2.7×10 ⁻⁸
	QTF [9]	1.53	370	5	21	1.05×10 ⁻⁴	/
CO ₂	LN-MFP	2	1900	10.6	350	3.71×10 ⁻³	2.53×10 ⁻⁸
	QTF [10]	2	365	5	2640	1.32×10 ⁻²	2.5×10 ⁻⁸
CH ₄	LN-MFP	3.3	1200	6.5	2.5	1.63×10 ⁻⁵	8.36×10 ⁻⁹
	QTF [11]	3.3	1	/	50	/	2.9×10 ⁻⁹
NH ₃	LN-MFP	9.7	1100	82.3	0.35	2.88×10 ⁻⁵	6.58×10 ⁻¹⁰

NNEA, normalized noise equivalent absorption coefficient; LN-MFP: lithium niobate multi-functional platform; IT: Integration time; QTF: quartz tuning fork; MDL: minimum detection limit; RMDL: relative minimum detection limit

LN-MFP's competitive photoacoustic performance arises from several complementary physical and practical factors. First, single-crystal lithium niobate exhibits substantially higher effective piezoelectric coupling (for appropriate crystal orientations and tensor components) than α -quartz, enabling LiNbO₃ devices to generate stronger electrical signals—and thus higher signal-to-noise ratios—for a given mechanical strain [5]. Second, the device geometry and resonant frequency were carefully optimized to balance acoustic coupling, thermoelastic drive and readout bandwidths ensuring robust multimodal transduction even for slowly relaxing species such as CO₂. Finally, lithium niobate's mature and rapidly evolving photonics ecosystem, featuring broad optical transparency, strong electro-optic properties and wafer-level processing, facilitates the integration of on-chip optics, waveguides and micro-collimation components that are challenging to implement in conventional quartz tuning-fork systems [6]. The chip-scale monolithic LN-MFP also minimizes parasitic capacitance and interconnect losses, enabling compact low-noise electronics near the transducer, thereby lowering the noise floor and enhancing NNEA.

Table 3s Photodetector using LN-MFP and pyroelectric sensors.				
Detector type	Characterized wavelength range	NEP (W/√Hz)	Response time	Responsivity (V/W)
LN-MFP	450–9770 nm	5.65×10^{-8}	1 s	373
QTF [12]	1540 nm	/	1 s	47.4
LiNbO ₃ [13]	/	1.623×10^{-7}	1.9 s	/
LiNbO ₃ [14]	8–11 μm	/	29 ms	/
LiTaO ₃ [15]	X-ray	5.02×10^{-8}	/	1800
Sb ₂ Te ₃ –Bi ₂ Te ₃ [16]	600–700 nm	8.0×10^{-9}	341 μs	38

LN-MFP: lithium niobate multi-functional platform; QTF: quartz tuning fork.

A concise comparison of the LN-MFP with representative pyroelectric and tuning-fork detectors is provided in Table 3s. In the current prototype the LN-MFP demonstrates high responsivity ($\sim 373 \text{ V W}^{-1}$) and a noise-equivalent power on the order of $5.6 \times 10^{-8} \text{ W Hz}^{-1/2}$; measured with a 1 s lock-in time constant (corresponding to $\sim 0.25 \text{ Hz}$ detection bandwidth). The device operates across a broad wavelength range from 450 nm to 9.77 μm and integrates PAS, LITES and direct photodetection functionalities on a single chip. The reported performance reflects a deliberate, system-level optimization, balancing geometry and readout to meet the combined requirements of all three sensing modalities. Although the LN-MFP is not necessarily optimized for any single modality, it uniquely combines high per-power responsivity, multimodal functionality and compatibility with wafer-scale photonics. Several straightforward engineering routes remain to further improve the absolute MDL, NEP and bandwidth, including the use of low-noise transimpedance amplifiers, micro-collimation or waveguide coupling to enhance optical power delivery, and tailored absorptive coatings at the fine base. These improvements will be explored in future work to further enhance performance while preserving the platform's integration advantages.

Table 4s LITES detection of trace gases using LN-MFP and QTF.						
Target analyte	Detector type	Wavelength (μm)	IT (s)	AL (cm)	MDL (ppm)	RMDL ($\text{ppm}\cdot\text{m}$)
NO ₂	LN-MFP	0.45	450	2.5	1	2.5×10^{-2}
H ₂ O	LN-MFP	1.39	1600	2.5	2.5	6.25×10^{-2}
	QTF [17]	1.39	289	50	1.4	7×10^{-1}
C ₂ H ₂	LN-MFP	1.53	1500	2.5	0.45	1.13×10^{-2}
	QTF [18]	1.53	70	20	0.36	7.2×10^{-2}
CO ₂	LN-MFP	2	1200	2.5	35	8.75×10^{-1}
	QTF [19]	2	100	100	20	2×10^1
CH ₄	LN-MFP	3.3	1900	2.5	0.9	2.25×10^{-2}
	QTF [20]	1.65	206	420	0.0035	1.47×10^{-2}
NH ₃	LN-MFP	9.7	1200	2.5	1.2	3×10^{-2}

AL: Absorption Length; QTF: quartz tuning fork; LN-MFP: lithium niobate multi-functional platform; MDL: minimum detection limit; RMDL: relative minimum detection limit; IT: Integration time;

For LITES, the key parameters include the laser wavelength, absorption length (AL), and integration time, as summarized in Table 4s. Since the MDL is directly proportional to the AL, the comparison includes not only the MDL but also the relative MDL for a fair and comprehensive evaluation. Table 4s shows that the LN-MFP attains superior RMDL for most channels (H₂O, C₂H₂, CO₂, NO₂, NH₃), with CH₄ as the only minor exception. This advantage stems from three factors: larger effective piezoelectric coupling of Y+128° LiNbO₃, an electrode layout that maximizes charge collection at regions of peak modal strain, and a resonance (~10.5 kHz) deliberately tuned to enhance thermoelastic/acoustic energy build-up for slowly relaxing species.

Reference

- Yue, W. et al. Crystal orientation dependence of piezoelectric properties in LiNbO₃ and LiTaO₃. *Opt. Mater.* **23**, 403-408 (2003).
- Boes, A. et al. Lithium niobate photonics: Unlocking the electromagnetic spectrum. *Science* **379**, eabj4396 (2023).
- Breitegger, P. et al. Towards low-cost QEPAS sensors for nitrogen dioxide detection. *Photoacoustics* **18**, 100169 (2020).

8. Rousseau R. et al. Monolithic double resonator for quartz enhanced photoacoustic spectroscopy. *Appl. Sci.* **11**: 2094 (2021).
9. Wang, R., Qiao, S., He, Y., & Ma, Y. Highly sensitive laser spectroscopy sensing based on a novel four-prong quartz tuning fork. *Opto-Electron. Adv.* **8**, 240275-1 (2025).
10. Zhang, Y. et al. Continuous real-time monitoring of carbon dioxide emitted from human skin by quartz-enhanced photoacoustic spectroscopy. *Photoacoustics* **30**, 100488 (2023).
11. Wu H. et al. Atmospheric CH₄ measurement near a landfill using an ICL-based QEPAS sensor with VT relaxation self-calibration. *Sens. Actuator B-Chem.* **297** 126753 (2019).
12. Zhou, S. et al. Realization of a infrared detector free of bandwidth limit based on quartz crystal tuning fork. *Opt. Laser Technol.* **113**, 261-265 (2019).
13. Aleks, M. et al. An overview of microelectronic infrared pyroelectric detector. *Eng. Sci.* **16**, 82-89 (2021).
14. Suen, J. Y. et al. Multifunctional metamaterial pyroelectric infrared detectors. *Optica* **4**, 276-279 (2017).
15. Kane, S. R. et al. Characterizing pyroelectric detectors for quantitative synchrotron radiation measurements. *Sensor. Actuat. A-Phys.* **387**, 116406 (2025).
16. Mauser, K. W. et al. Resonant thermoelectric nanophotonics. *Nat. Nanotechnol.* **12**, 770-775 (2017).
17. Xu, L. et al. Multigas sensing technique based on quartz crystal tuning fork-enhanced laser spectroscopy. *Anal. Chem.* **92** 14153-14163 (2020).
18. He, Y. et al. Hydrogen-enhanced light-induced thermoelastic spectroscopy sensing. *Photonics Res.* **13** 194-200 (2024).
19. Bojęś, P. et al. Dual-band light-induced thermoelastic spectroscopy utilizing an antiresonant hollow-core fiber-based gas absorption cell. *Appl. Phys. B* **129**, 177 (2023).
20. Shang, Z. et al. Robust and compact light-induced thermoelastic sensor for atmospheric methane detection based on a vacuum-sealed subminiature tuning fork. *Photoacoustics* **42**, 100691 (2025).”

Response to reviewers' comments

Dear reviewers:

Please find enclosed our answers to the three referees' comments. We have addressed all questions raised by both reviewers, and revised our manuscript, accordingly.

Please Note: reviewers' comments are in black; our comments are in *red italic*; the original paper text is in red and revised text is in blue.

Reviewer #1:

The authors have addressed my previous comments well overall, and the quality of the revised manuscript has improved. However, I still have several remaining concerns.

1. While the multifunctional operation of the LN-MFP platform is interesting, the level of novelty remains limited. The individual sensing mechanisms (PAS, LITES, and thermoelastic photodetection) have been demonstrated previously using QTF-based and MEMS-based systems, and the performance improvements presented here appear incremental rather than introducing a fundamentally new advancement. The manuscript should clearly describe the specific scientific or technological contribution that is unique to this LN-based approach, and how it advances the current state of the art. I believe that strengthening this aspect is very important for meeting the expectations of Nature Communications.

We thank the reviewer for this important comment. We fully agree that it is essential to clearly articulate what is unique about the present LN-based approach. Our key point is that this work does not simply transplant known mechanisms onto a different substrate instead, it establishes a new material and system-level platform in which these mechanisms are unified and exploited in ways that are not attainable with quartz or conventional MEMS technologies.

*The most fundamental distinction between lithium niobate (LN) and existing technologies lies in its intrinsic multifunctionality and strong compatibility with on-chip integration. LN exhibits significantly richer multi-physical-field coupling—encompassing piezoelectric, pyroelectric, electro-optic, and photoelastic effects—than quartz. In recent years, extensive studies on LN have enabled a broad range of optoelectronic components, including light sources [Nature **615**, 411-417 (2023)], optical waveguides [Nat. Photon. **18**, 218-223 (2024)], optical isolators [Nat. Photon. **17**, 666-671 (2023)], and optical modulators [Nat. Commun. **16**, 2281 (2025)]. Although these technologies target applications in optical communication or sensing, truly practical LN-based sensing applications remain scarce.*

In this work, we take the lead in fabricating devices directly on LN wafers and successfully demonstrate multiple sensing-related functionalities spanning the visible to the mid-infrared spectral range. Moreover, we integrate a mid-infrared quantum cascade laser (QCL) with the LN multi-functional platform and validate its capability for gas sensing. Therefore, this contribution of this work should be viewed as a system-level innovation rather than an isolated device-level demonstration.

In the revised manuscript, we now highlight four aspects of novelty:

(1) Material-level novelty: intrinsically multifunctional LN platform.

The most fundamental difference between lithium niobate (LN) and existing quartz- or Si-based technologies is LN's intrinsic multifunctionality and strong multi-physical-field coupling. LN simultaneously exhibits strong piezoelectric, pyroelectric, electro-optic, and photoelastic effects, whereas quartz provides only comparatively weak piezoelectric coupling and lacks electro-optic and pyroelectric responses. Over the past years, these material properties have enabled a rich ecosystem of LN-based optoelectronic components, including light sources, optical waveguides, optical isolators, and optical modulators [Nature 615, 411–417 (2023); Nat. Photon. 18, 218–223 (2024); Nat. Photon. 17, 666–671 (2023); Nat. Commun. 16, 2281 (2025)]. However, despite this progress, practical and fully integrated LN-based sensing implementations remain scarce. Our work fills this gap by realizing a multifunctional sensing platform directly on LN wafers and by demonstrating that LN can serve as the core material for a unified PAS/LITES/photodetection platform spanning the visible to the mid-infrared.

(2) Fundamental material properties and a design paradigm distinct from quartz.

The fabrication processes and intrinsic physical properties of LN are fundamentally different from those of quartz. LN is a ferroelectric crystal with spontaneous polarization and belongs to the 3m point group, whereas quartz is non-ferroelectric (point group 32) and lacks spontaneous polarization. As a result, LN exhibits piezoelectric coefficients and electromechanical coupling factors more than an order of magnitude higher than those of quartz (e.g., $d_{22} \approx 25$ pC/N for LN versus $d_{11} \approx 2.3$ pC/N for quartz), and additionally offers strong electro-optic, nonlinear-optical, and pyroelectric effects that quartz completely lacks [Nat. Commun. 14, 4856 (2023)]. Because of these fundamental differences in crystal symmetry and ferroelectric nature, LN-based sensors cannot be designed by simply applying “scaled-down” quartz-QTF paradigms. The fork-shaped resonator geometry, electrode layout, and fabrication flow used in this work are specifically engineered to exploit LN's unique multi-physics response. These details were fully demonstrated in the previous revision. It is precisely these new design and manufacturing strategies that enable the superior multimodal performance of the LN-MFP platform presented here.

(3) Mechanism-level advances for PAS, photodetection, and LITES on LN.

(a) Photoacoustic spectroscopy (PAS).

A defining characteristic of PAS is that the signal amplitude scales inversely with the interaction volume. By developing an on-chip–integrable PAS transducer based on LN, we can simultaneously leverage (i) the strong PAS scaling at small volumes and (ii) the inherent capability of a wafer-scale photonic platform. LN wafers can be micro-fabricated and co-integrated with optoelectronic or MEMS subsystems to realize compact PAS sensors, which is extremely challenging with quartz QTFs. Moreover, LN's much larger piezoelectric coefficients and electromechanical coupling factors translate directly into higher acoustic-to-electric conversion efficiency: under the same acoustic pressure, LN generates a substantially larger electrical output than quartz. LN's relatively high dielectric permittivity ($\epsilon_{11} \approx 85$, $\epsilon_{33} \approx 30$) further enables a strong polarization response under high-frequency and high-field operation. These material advantages provide a fundamentally different and improved operating regime for PAS compared with QTF-based sensors.

(b) Thermoelastic photodetection.

LN exhibits strong electro-optic and nonlinear-optical properties that quartz fundamentally lacks. As a non-centrosymmetric ferroelectric crystal, LiNbO_3 supports a strong linear electro-optic (Pockels) effect and pronounced photorefractive and pyroelectric responses, enabling direct optical-to-electrical energy conversion and efficient optical phase/intensity modulation. Quartz, by contrast, does not exhibit second-order nonlinearity and is limited to weak piezo-photoelastic effects, making it unsuitable for high-performance thermoelastic photodetection. Furthermore, LN devices can operate with ultrafast response and extremely wide bandwidths, reaching the GHz–THz regime [Nat. Commun. 16, 8864 (2025)]. When exploited as photothermal or thermoelastic photodetectors, LN devices benefit simultaneously from thermoelastic and Joule-heating effects, setting a much higher intrinsic ceiling on sensitivity and bandwidth than quartz- or Si-based detectors.

(c) Light-induced thermoelastic spectroscopy (LITES).

In LITES, the same LN-MFP serves as the multi-physics transducer. When LN is used as both optical modulation and detection medium, several performance enhancements become available. LN's broad electro-optic bandwidth allows rapid laser frequency modulation and improved frequency stability, enhancing both measurement speed and spectral resolution. LN-based devices also naturally support phase-sensitive detection schemes, enabling precise extraction of both amplitude and phase of the absorption signal. Because LN provides a single material platform for both high-speed modulators and sensitive detectors, future LN-based LITES systems can integrate modulator and detector within the same material, enabling high-frequency modulation and lock-in detection in a unified architecture. Such co-integration is not achievable with quartz-based or conventional MEMS sensors and can significantly reduce the detection limits while expanding dynamic range [Nat. Commun. 16, 8864 (2025)]. Additionally, LN's high Curie temperature ($T_c \approx 1210^\circ\text{C}$), large pyroelectric coefficient ($p \approx 40 \mu\text{C m}^{-2} \text{K}^{-1}$), and excellent thermal robustness allow stable operation over wide temperature ranges and under high optical intensity, making LN particularly suitable for harsh-environment sensing (e.g., combustion diagnostics) beyond the reach of most QTF- or MEMS devices.

(4) System-level innovation and compatibility with the LN photonics ecosystem.

As summarized in the new Table 1r of the revised manuscript, recent works have established LN as a leading platform for integrated light sources, frequency combs, high-speed modulators, and wavelength converters spanning the visible to the mid-infrared [Science 379, eabj4396 (2023) and related references]. These studies repeatedly highlight the potential of LN integrated photonics for spectroscopy, but a multifunctional sensing “front end” compatible with this ecosystem has been missing. Our work addresses this gap by introducing an LN-based multifunctional platform (LN-MFP) that unifies photoacoustic spectroscopy, light-induced thermoelastic spectroscopy, thermoelastic photodetection and, at the system level, a mid-infrared QCL source within a single hybrid module. The sensor is fabricated using commercially available LN wafers, assembled into a PCB-level co-packaged LN-MFP–QCL module, and experimentally validated for gas sensing. This demonstrates that the proposed LN platform is not only physically distinct from quartz QTFs and traditional MEMS sensors, but also technologically compatible with the existing LN integrated-photonics infrastructure,

thereby enabling future “all-lithium-niobate” spectroscopic chips.

Table 1r Overview of state-of-the-art lithium niobate integrated photonic devices.		
Article	Device	Ref.
Ultrafast tunable photonic-integrated extended-DBR Pockels laser	Light Source	[1]
Programmable multifunctional integrated microwave photonic circuit on thin-film lithium niobate	Modulator	[2]
Broadband electro-optic frequency comb generation in a lithium niobate microring resonator	Light Source	[3]
Cascaded nonlinear down-conversion in poling-free lithium niobate nanophotonic waveguides	Light Source	[4]
Sub-1 Volt and high-bandwidth visible to near-infrared electro-optic modulators	Modulator	[5]
Ultrafast tunable lasers using lithium niobate integrated photonics	Light Source	[6]
Integrated lithium niobate photonics for sub-ångström snapshot spectroscopy	Spectrometer	[7]
Visible-to-ultraviolet frequency comb generation in lithium niobate nanophotonic waveguides	Light Source	[8]
Versatile Lithium Niobate Platform for Photoacoustic/Thermoelastic Gas Sensing and Photodetection	Platform (integrated source and detector)	this work

In summary, the originality of this work lies in establishing a lithium-niobate-based multifunctional platform in which three fundamentally different sensing mechanisms, photoacoustic sensing, thermoelastic photodetection, and light-induced thermoelastic sensing, together with a mid-infrared QCL source are realized within a single, strongly multi-physics-coupled LN architecture and validated in a compact PCB-level module. This creates a complete and coherent progression from material properties to device concept and ultimately to a practically deployable hybrid sensing system. Thereby defining a new multimodal sensing framework tailored to the LN integrated photonics ecosystem.

We believe that this material-level and system-level advance go beyond incremental performance improvements and is well aligned with the expectations of Nature Communications for interdisciplinary work that opens genuinely new research directions. To

make these points clearer to the reader, we have revised the Introduction and Discussion sections to more explicitly highlight these unique scientific and technological contributions. We have also added Table 1r to situate our work within the broader context of state-of-the-art LN integrated photonic devices.

References

- [1] Siddharth A. et al. Ultrafast tunable photonic-integrated extended-DBR Pockels laser. *Nat. Photon.* **19**, 709-717 (2025).
- [2] Wei C. et al. Programmable multifunctional integrated microwave photonic circuit on thin-film lithium niobate. *Nat. Commun.* **16**, 2281 (2025).
- [3] Zhang M. et al. Broadband electro-optic frequency comb generation in a lithium niobate microring resonator. *Nature* **568**, 373-377 (2019).
- [4] Yan C. et al. Cascaded nonlinear down-conversion in poling-free lithium niobate nanophotonic waveguides. *Nat. Commun.* **16**, 9987.
- [5] Renaud D. et al. Sub-1 Volt and high-bandwidth visible to near-infrared electro-optic modulators. *Nat. Commun.* **14**, 1496 (2023).
- [6] Snigirev V. et al. Ultrafast tunable lasers using lithium niobate integrated photonics. *Nature* **615**, 411-417 (2023).
- [7] Yao Z. et al. Integrated lithium niobate photonics for sub-ångström snapshot spectroscopy. *Nature* **646**, 567-575 (2025).
- [8] Wu T. H. et al. Visible-to-ultraviolet frequency comb generation in lithium niobate nanophotonic waveguides. *Nat. Photon.* **18**, 218-223 (2024).

In the abstract replace the text:

“By leveraging LN’s strong piezoelectric and thermoelastic properties, along with an optimized fork-shaped design, our platform achieves high sensitivity and selectivity across a broad spectral range—from visible to long-wave infrared wavelengths.”

with the following:

“Leveraging the intrinsic multi-physics nature of ferroelectric lithium niobate, the proposed LN multi-functional platform (LN-MFP) unifies photoacoustic spectroscopy, light-induced thermoelastic spectroscopy, and thermoelastic photodetection within a single chip, while operating over a broad spectral range spanning from the visible to the mid-infrared.”

And substitute the abstract text:

“This compact, integrated, multi-functional architecture significantly reduces system complexity and footprint, paving the way for portable, scalable sensing systems in environmental monitoring, point-of-care diagnostics, and on-site chemical analysis, marking a key advancement in integrated photonics.”

with:

“This compact, hybrid, multi-functional architecture significantly reduces system complexity

and footprint compared with conventional benchtop systems and is intrinsically compatible with the rapidly developing lithium niobate integrated photonics ecosystem. The LN-MFP establishes a lithium niobate-based multimodal sensing framework and provides a core sensing building block for future all-lithium-niobate spectroscopic chips for environmental monitoring, point-of-care diagnostics, and on-site chemical analysis.”

Add the following paragraph in the main text Page 4 Line 104 at the beginning of the section:

Design and Fabrication of the Lithium Niobate Multi-Functional Platform

“The design philosophy of the LN-MFP fundamentally differs from that of conventional quartz- or Si-based PAS/LITES devices. Lithium niobate is a ferroelectric crystal with 3m point-group symmetry and spontaneous polarization, leading to piezoelectric coefficients and electromechanical coupling factors more than one order of magnitude larger than those of quartz (for example, $d_{22} \approx 25$ pC/N for LN versus $d_{11} \approx 2.3$ pC/N for quartz). At the same time, LN exhibits strong electro-optic, nonlinear-optical, and pyroelectric effects that are absent in quartz and most MEMS materials. These intrinsic properties mean that LN-based sensors cannot be obtained by simply scaling or copying established quartz tuning fork designs; instead, the fork geometry, electrode layout, and fabrication process in this work are specifically engineered to harness LN’s multi-physics coupling. As a result, a single LN-MFP chip can simultaneously operate as a high-efficiency acoustic transducer for PAS, a thermoelastic detector, and a platform for LITES, thereby enabling multimodal operation that is not available in conventional quartz or MEMS devices.”

Replace the following text on main article Page 20 Line 429:

“The development of the lithium niobate multi-functional platform represents a significant breakthrough in integrated spectroscopic sensing technologies. By harnessing the multifunctional properties of LN and incorporating a resonant fork-shaped design, we have successfully integrated photoacoustic spectroscopy, light induced thermoelastic spectroscopy, and photodetection into a single compact device. Experimental results reveal high sensitivity and selectivity across a broad spectral range, achieving detection limits suitable for real-world applications in environmental monitoring and industrial process control. This platform effectively addresses the integration challenges associated with conventional spectroscopic systems, providing a scalable and versatile solution that enhances performance while significantly reducing size and complexity, and system requirements.”

with:

“Furthermore, by co-packaging the LN-MFP chip with a mid-infrared QCL chip and the associated drive and readout electronics on a compact PCB, we demonstrate that the LN-MFP can be seamlessly interfaced with semiconductor laser sources using standard microelectronic assembly techniques. Experimental results reveal high sensitivity and selectivity across a broad spectral range, achieving detection limits suitable for real-world applications in environmental monitoring and industrial process control. Even though the current implementation is at the PCB level rather than monolithic, this hybrid QCL–LN architecture already reduces the

footprint and alignment complexity of the spectrometer and establishes the LN-MFP as a practical building block for future fully integrated lithium-niobate spectroscopic chips.

In summary, we have established a lithium-niobate-based multi-functional platform (LN-MFP) that unifies three fundamentally different sensing mechanisms, photoacoustic spectroscopy, light-induced thermoelastic spectroscopy, and thermoelastic photodetection, within a single ferroelectric crystal chip. The LN-MFP exploits the strong piezoelectric, pyroelectric, electro-optic, and photoelastic responses of lithium niobate to achieve efficient acoustic-to-electric and thermoelastic transduction, while operating over a spectral range extending from the visible to the mid-infrared. Beyond the device level, we demonstrate a PCB-level co-packaged LN-MFP–QCL module capable of mid-infrared gas sensing, thereby validating that the LN-MFP can serve as a practical sensing front-end compatible with semiconductor laser sources and standard microelectronic assembly. Together with recent progress in LN-based lasers, frequency combs, and high-speed modulators, this multifunctional sensing platform fills a key gap in the LN photonics landscape and points toward future “all-lithium-niobate” spectroscopic chips in which light generation, modulation, and multimodal detection are realized within a single material system. We therefore view the main advance of this work not as an incremental improvement in a single performance metric, but as the introduction of a new LN-centric material and system-level paradigm for integrated spectroscopic sensing.”

Additionally, the term “on-chip integration”, especially in the abstract, may lead readers the impression of monolithic or wafer-scale integration. In this work, however, the demonstrated QCL-LN configuration is based on PCB-level co-packaging rather than true chip-scale integration. I recommend clarifying this distinction in both the abstract and the main text to avoid overstating the integration level. In addition, it would be helpful for the authors to more explicitly describe what the key novelty and strengths of this work are, although the integration remains at the PCB level. Providing a clear justification of the scientific value and technological impact of this work, despite not achieving on-chip integration, would strengthen the manuscript’s positioning.

We thank the reviewer for this important comment and agree that our present implementation corresponds to PCB-level co-packaging of a discrete QCL chip with the LN-MFP chip, rather than to monolithic or wafer-scale integration. In the original submission we used the expressions “on-chip integration” and “on-chip sensor” to emphasize that the sensing functions (photoacoustic spectroscopy, light-induced thermoelastic spectroscopy, and photodetection) are monolithically implemented on the lithium niobate chip. However, as the reviewer correctly points out, this wording may inadvertently give the impression that the QCL source is also already integrated on the same chip.

To avoid any possible overstatement of the integration level, we have carefully revised the terminology in both the abstract and the main text:

In addition, following the reviewer’s suggestion, we now more clearly articulate the novelty and strengths of the work despite the current integration level:

- First, at the device level, the LN-MFP monolithically integrates photoacoustic spectroscopy, light-induced thermoelastic spectroscopy, and photodetection on a*

single lithium-niobate chip and operates over a broad spectral range from the visible to the long-wave infrared. This multifunctionality and bandwidth, enabled by the intrinsic material properties of LN and the fork-shaped resonant design, remain the central scientific contribution of the work.

- *Second, at the system level, the revised text emphasizes that co-packaging the LN-MFP chip with a mid-infrared QCL chip and on-board drive/readout electronics on a compact PCB demonstrates a realistic, industry-compatible hybrid architecture. Even though the light source is not yet monolithically integrated, this PCB-level QCL–LN module already reduces system footprint and alignment complexity compared with conventional benchtop spectrometers and validates the LN-MFP as a practical building block for future fully integrated lithium-niobate spectroscopic chips.*

To make these points explicit to the reader, we have (i) revised the last part of the abstract and (ii) added clarifying sentences to the concluding paragraph of the main text that highlight the scientific value and technological impact of the current PCB-level implementation while clearly positioning it as an intermediate, but important, step toward monolithic QCL–LN integration.

The manuscript revisions are listed below.

In the abstract we replace the following sentences:

“To further enhance on-chip integration, we implement a custom packaging solution featuring a printed circuit board with transimpedance amplification and a quantum cascade laser chip, enabling direct wire bonding to the LN-MFP. Exploiting a 4.6 μm quantum cascade laser chip, we successfully demonstrate carbon monoxide detection via second-harmonic measurements, highlighting the LN-MFP’s potential for fully integrated spectroscopic applications. This compact, integrated, multi-functional architecture significantly reduces system complexity and footprint, paving the way for portable, scalable sensing systems in environmental monitoring, point-of-care diagnostics, and on-site chemical analysis, marking a key advancement in integrated photonics.”

with the text:

“To demonstrate system-level integration in a practical form factor, we implement a custom packaging solution in which the LN-MFP chip and a 4.6 μm quantum cascade laser (QCL) chip are co-mounted on a printed circuit board together with transimpedance amplification, forming a hybrid PCB-level QCL–LN module with direct wire bonding to the LN-MFP. Using this co-packaged module, we successfully demonstrate carbon monoxide detection via second-harmonic measurements, thereby validating that the LN-MFP can operate as the core sensing element of a compact hybrid QCL–LN spectroscopic system and outlining a clear route toward future fully integrated on-chip implementations. This compact, hybrid, multi-functional architecture significantly reduces system complexity and footprint compared with conventional benchtop systems and is intrinsically compatible with the rapidly developing lithium niobate integrated photonics ecosystem. The LN-MFP establishes a lithium niobate–based multimodal sensing framework and provides a core sensing building block for future all-lithium-niobate spectroscopic chips for environmental monitoring, point-of-care diagnostics, and on-site

chemical analysis.”

We changed the subsection title of main article on Page 18 Line 401:

“On-chip LN-MFP Sensor”

to

“PCB-level Co-packaged LN-MFP Sensor Module”

On Page 18 Line 402 we substitute the following text:

“Based on the abovementioned multifunctional capabilities, we introduced an designed on-chip packaging approach and evaluated its performance. As the photograph shown in Fig. 5a,”

with the following text:

“Based on the abovementioned multifunctional capabilities, we next implemented a PCB-level co-packaged LN-MFP sensor module and evaluated its performance. As shown in the photograph in Fig. 5a,

On Page 19 Line 424 we substitute the following text:

“Figure 5d presents the $2f$ signal of 8000 ppm CO in N₂ measured using the packaged LN-MFP-QCL chip, demonstrating the platform’s strong potential for fully integrated on-chip spectroscopic applications.”

with the text:

“Figure 5d presents the $2f$ signal of 8,000 ppm CO in N₂ measured using the PCB-level co-packaged LN-MFP-QCL module, demonstrating that the LN-MFP can serve as a compact and efficient sensing element in a hybrid QCL-LN configuration and highlighting a realistic pathway toward future fully integrated on-chip spectroscopic systems.”

Replaced Figure 5 caption:

“Fig. 5 | On chip LN-MFP sensor. a Photograph of the packaged on-chip sensor. b Optical-microscope image of the coupling region showing the QCL facet and the LN-MFP prong gap. c Schematic diagram of the free-space optical coupling. d $2f$ photoacoustic spectrum recorded from 8,000 ppm CO in N₂.”

with:

“Fig. 5 | PCB-level co-packaged LN-MFP-QCL sensor module. a Photograph of the packaged PCB-level hybrid sensor module. b Optical-microscope image of the coupling region showing the QCL facet and the LN-MFP prong gap. c Schematic diagram of the free-space optical coupling. d $2f$ photoacoustic spectrum recorded from 8,000 ppm CO in N₂.”

In the main article Page 20 Line 429, we replaced the following text:

“The development of the lithium niobate multi-functional platform represents a significant breakthrough in integrated spectroscopic sensing technologies. By harnessing the multifunctional properties of LN and incorporating a resonant fork-shaped design, we have successfully integrated photoacoustic spectroscopy, light induced thermoelastic spectroscopy, and photodetection into a single compact device. Experimental results reveal high sensitivity and selectivity across a broad spectral range, achieving detection limits suitable for real-world applications in environmental monitoring and industrial process control. This platform effectively addresses the integration challenges associated with conventional spectroscopic systems, providing a scalable and versatile solution that enhances performance while significantly reducing size and complexity, and system requirements.”

with:

“Furthermore, by co-packaging the LN-MFP chip with a mid-infrared QCL chip and the associated drive and readout electronics on a compact PCB, we demonstrate that the LN-MFP can be seamlessly interfaced with semiconductor laser sources using standard microelectronic assembly techniques. Experimental results reveal high sensitivity and selectivity across a broad spectral range, achieving detection limits suitable for real-world applications in environmental monitoring and industrial process control. Even though the current implementation is at the PCB level rather than monolithic, this hybrid QCL–LN architecture already reduces the footprint and alignment complexity of the spectrometer and establishes the LN-MFP as a practical building block for future fully integrated lithium-niobate spectroscopic chips.”

2. Although the authors provide additional details regarding the free-space alignment between the QCL and the LN-MFP, the optical coupling efficiency is still not reported. Since key performance metrics such as NNEA and MDL depend on the optical power actually delivered to the sensing region, the absence of a coupling-efficiency estimate introduces uncertainty in the quantitative interpretation of the results. Please clarify how the NNEA values were calculated rigorously.

We thank the reviewer for raising this important point regarding optical coupling efficiency. Based on the structural parameters provided in both the initial manuscript and the previous revision, we have now conducted an optical simulation using Zemax to estimate the coupling efficiency from the QCL chip to the sensing region between the LN-MFP prongs.

In the simulation, we assume a conical Gaussian beam emission from the QCL chip, with a divergence angle of $\sim 70^\circ$. The chip to LN-MFP distance is approximately $100\ \mu\text{m}$, as shown in the Fig. 1r a. While the lateral divergence is constrained by the gap between the two prongs of the LN-MFP, the vertical divergence extends beyond the LN-MFP plane ($500\ \mu\text{m}$ thickness), resulting in partial optical loss as shown in the Fig. 1r b. This out-of-plane component does not contribute to the detection process. Based on this configuration, Zemax modeling estimates the effective optical coupling efficiency to be $\sim 29\%$.

Fig. 1r | Optical coupling between the QCL chip and the LN-MFP. a Schematic diagram of the free-space optical coupling. **b** Optical simulation by ZEMAX.

Therefore, we have added the following clarification to the revised manuscript on Page 19, Line 420:

“The optical coupling efficiency was estimated to be ~29%.”

As correctly noted by the reviewer, the effective optical power used in the NNEA calculation corresponds to this coupled power rather than the total QCL output. To clarify this, we have included a rigorous explanation of the NNEA evaluation in the revised Supplementary Section 9, Page 13 Line 294:

“9. The calculation of NNEA

To quantitatively assess the performance of the LN-MFP-based spectroscopic sensor, the normalized noise-equivalent absorption (NNEA) was evaluated for the different gas channels. The NNEA was calculated according to the following expression:

$$\text{NNEA} = \frac{\alpha_{\min} P_{\text{eff}}}{\sqrt{\text{ENBW}}} \quad (2)$$

where P_{eff} is the effective optical power delivered to the sensing region, α_{min} denotes the minimum detectable absorption coefficient, and $ENBW$ is the equivalent noise bandwidth of the detection system. The ENBW was determined by the settings of the lock-in amplifier. In our experiments, a time constant of 1 s and a filter slope of 12 dB/oct resulted in an equivalent noise bandwidth of 0.25 Hz.”

We hope this detailed response and new supplementary material fully address the reviewer's concern. Many thanks for the great improvement again.

Reviewer #2:

The authors have made significant revision in the manuscript and well answered most of my questions. Here I have only a few minor comments for authors' consideration and clarification. (1) I appreciate the authors added N₂ background signals in Fig. 2 for direct comparison, but there is significant non-zero signal in (b) and (c). Please explain and mention whether it is an issue for applications.

We thank the reviewer for this insightful comment. The baseline may arise from system-induced offsets, including the residual electronic background of the lock-in amplifier, imperfections in laser intensity modulation, and thermo-mechanical background excitation of the LN-MFP at resonance. However, this baseline offset does not affect the actual sensing performance or practical operation, as the measurements rely on differential 2f signal amplitudes relative to the baseline. As long as the baseline remains stable over time, even if it is non-zero, it can be effectively compensated through standard data processing and baseline-correction procedures.

(2) The authors made detailed response to the previous question Q3 regarding comparison to LITES. I notice this part has been added to supplemental, but nothing is mentioned in the main text. Please inform readers this part is discussed in the supplemental information.

Following this comment, the following sentence has been added to the main article Page 18 Line 385:

“A comparison with state-of-the-art LITES is provided in the Supplementary Section 10.”

(3) In the previous question Q5, the Table 3s compares pyroelectric detectors. Is it possible to include photovoltaic detector in the table as well, considering its wide use in spectroscopy.

Thanks for the constructive suggestion. The photovoltaic detectors were added to the revised Table 3s of the supplementary Page 15 Line 326:

Table 3s Photodetector using LN-MFP and pyroelectric/photovoltaic sensors.					
Detector type	Category	Characterized wavelength range	NEP (W/ $\sqrt{\text{Hz}}$)	Response time	Responsivity (V/W)
LN-MFP	Pyroelectric	450–9770 nm	5.65×10^{-8}	1 s	373
QTF [12]	Pyroelectric	1540 nm	/	1 s	47.4
LiNbO ₃ [13]	Pyroelectric	/	1.623×10^{-7}	1.9 s	/
LiNbO ₃ [14]	Pyroelectric	8–11 μm	/	29 ms	/
LiTaO ₃ [15]	Pyroelectric	X-ray	5.02×10^{-8}	/	1800
Sb ₂ Te ₃ –Bi ₂ Te ₃ [16]	Pyroelectric	600–700 nm	8.0×10^{-9}	341 μs	38
HgCdTe [17]	Photovoltaic	4.8 μm	1.7×10^{-13}	13 ns	/
Graphene [18]	Photovoltaic	1550 nm	4.54×10^{-12}	/	210

LN-MFP: lithium niobate multi-functional platform; QTF: quartz tuning fork.

[17] Wang, Y. et al. Fast uncooled mid-wavelength infrared photodetectors with heterostructures of van der Waals on epitaxial HgCdTe. *Adv. Mater.* **34**, 2107772 (2022).

[18] Molaei-Yeznabad, A. & Abedi, K. Optimal design of graphene-based plasmonic enhanced photodetector using PSO. *Sci. Rep.* **14**, 15291 (2024).

Reviewer #3:

The authors have carefully revised the manuscript according to my comments. I am content with the revisions and suggesting acceptance now.

Many thanks for the great improvement.